# Is Curcumin Intake Really Effective for Chronic Inflammatory Metabolic Disease? A Review of Meta-Analyses of Randomized Controlled Trials

**DOI:** 10.3390/nu16111728

**Published:** 2024-05-31

**Authors:** Young-Min Lee, Yoona Kim

**Affiliations:** 1Department of Practical Science Education, Gyeongin National University of Education, Gyesan-ro 62, Gyeyang-gu, Incheon 21044, Republic of Korea; ymlee@ginue.ac.kr; 2Department of Food and Nutrition, Institute of Agriculture and Life Science, Gyeongsang National University, Jinju 52828, Republic of Korea

**Keywords:** curcumin, meta-analysis, randomized controlled trials, chronic inflammatory metabolic disease

## Abstract

This review aimed to examine the effects of curcumin on chronic inflammatory metabolic disease by extensively evaluating meta-analyses of randomized controlled trials (RCTs). We performed a literature search of meta-analyses of RCTs published in English in PubMed^®^/MEDLINE up to 31 July 2023. We identified 54 meta-analyses of curcumin RCTs for inflammation, antioxidant, glucose control, lipids, anthropometric parameters, blood pressure, endothelial function, depression, and cognitive function. A reduction in C-reactive protein (CRP) levels was observed in seven of ten meta-analyses of RCTs. In five of eight meta-analyses, curcumin intake significantly lowered interleukin 6 (IL-6) levels. In six of nine meta-analyses, curcumin intake significantly lowered tumor necrosis factor α (TNF-α) levels. In five of six meta-analyses, curcumin intake significantly lowered malondialdehyde (MDA) levels. In 14 of 15 meta-analyses, curcumin intake significantly reduced fasting blood glucose (FBG) levels. In 12 of 12 meta-analyses, curcumin intake significantly reduced homeostasis model assessment of insulin resistance (HOMA-IR). In seven of eight meta-analyses, curcumin intake significantly reduced glycated hemoglobin (HbA1c) levels. In eight of ten meta-analyses, curcumin intake significantly reduced insulin levels. In 14 of 19 meta-analyses, curcumin intake significantly reduced total cholesterol (TC) levels. Curcumin intake plays a protective effect on chronic inflammatory metabolic disease, possibly via improved levels of glucose homeostasis, MDA, TC, and inflammation (CRP, IL-6, TNF-α, and adiponectin). The safety and efficacy of curcumin as a natural product support the potential for the prevention and treatment of chronic inflammatory metabolic diseases.

## 1. Introduction

Chronic (non-communicable) diseases can be defined as long-term (3 months or more) health conditions that develop overtime [1]. Diabetes, cardiovascular disease, Alzheimer’s disease (AD), Parkinson’s disease, and cancer are included in chronic disease [1], which are major players of mortality responsible for 70% of all mortality worldwide [2]. Several contributors to chronic diseases are genetics, unhealthy dietary patterns, sedentary lifestyles, stress, alcohol intake, smoking, environmental pollutants, radiation exposure, infections, inflammation, and so on [3,4,5,6].

The development of chronic disease is especially associated with unregulated ongoing chronic inflammation conditions [7]. Chronic inflammation-related diseases include metabolic syndrome, type 2 diabetes, cardiovascular disease, non-alcoholic fatty liver disease (NAFLD), depression, and cognitive impairment [7,8]. Due to the relationship between oxidative stress and inflammation through activating transcription factors [9], anti-inflammatory activity, as well as antioxidant activity, could offer a benefit for the prevention and treatment of chronic inflammatory metabolic disease.

Turmeric is obtained from *Curcuma longa* L. (a tuberous herbaceous perennial plant, which has yellow flowers and broad leaves) and grows in tropical climates. Turmeric (*Curcuma longa*) is a curry spice and a herbal medicine that originated in India [10]. Turmeric as a yellow compound is widely used as food flavoring and coloring [11,12]. Turmeric comprises three bioactive curcuminoids [curcumin (2–5% of turmeric), demethoxycurcumin, and bisdemethoxycurcumin), protein, sugars, resins, and volatile oils (such as natlantone, turmerone, and zingiberene). In particular, curcumin (1,7-bis-(4-hydroxy-3-methoxyphenyl)-hepta-1,6-diene-3,5-dione, also called diferuloylmethane), is a major curcuminoid in the ginger family [13,14,15]. Curcumin is the main active component of *Curcuma longa*, which exerts antioxidant, anti-inflammatory, and anti-aging effects attributable to its lipophilic phenolic O-H and the CH-H hydrogen group [16,17,18,19]. The anti-inflammatory and antioxidant properties of curcumin can play a key role in the prevention and treatment of chronic inflammatory metabolic diseases [20,21].

The objective of this review was to clarify the effect of curcumin on chronic inflammatory metabolic disease by examining meta-analyses of randomized controlled trials (RCTs). The present study includes a meta-analysis of as many RCTs as possible to extensively investigate the effects of curcumin intake on chronic inflammatory metabolic diseases.

## 2. Materials and Methods

We investigated meta-analyses of the effects of curcumin on inflammatory diseases in human studies. A literature search was conducted in the PubMed^®^/MEDLINE^®^ (https://www.ncbi.nlm.nih.gov/pubmed/) (accessed on 7 August 2023) database and was restricted to full articles in English up to 31 July 2023. The search terms included meta-analysis combined with curcumin. Reference lists of selected papers were also examined. The titles and abstracts of articles were initially screened, and then full texts of the literature were reviewed for final study selection. Inclusion criteria were glycemic control, glucose, lipids, blood pressure (BP), inflammatory markers, type 2 diabetes mellitus (T2DM), hypertension, metabolic syndrome, non-alcoholic fatty liver disease (NAFLD), cardiovascular disease (CVD), endothelial function, flow-mediated dilation (FMD), obesity, body weight (BW), depression, and cognitive function. Meta-analyses of human studies were included, and non-English articles were excluded. Articles addressing skin, dermatitis, oral, periodontitis, gingivitis, quality of life, tumor, cancer, irritable bowel syndrome, bone, chronic obstructive pulmonary disease (COPD), coronavirus disease (COVID-19), nephropathy, colitis, arthritis, muscle, liver enzymes (function), and hemodialysis were excluded among the 212 articles searched with the terms curcumin and meta-analysis in PubMed^®^/MEDLINE^®^. Finally, fifty-four meta-analyses that specifically addressed these inclusion criteria were selected and included in this review. A flowchart of this study’s screening and selection process is reported in Figure 1.

## 3. Results

### 3.1. Identification and Selection of Studies

A total of 210 studies were initially identified, 209 from the database search and one from the manual search. The titles and abstracts of 208 studies were screened by two authors. Subsequently, 71 studies were excluded. The remaining 137 studies underwent double full-text review. Subsequently, 83 studies were excluded; 82 were excluded because of unrelated outcomes, and one non-English report was excluded. Finally, 54 studies were included in the systematic review. The Preferred Reporting Items for Systematic Reviews and Meta-Analyses (PRISMA) flowchart for the selection process is presented in Figure 1.

### 3.2. Inflammatory Markers

The influence of curcumin on inflammatory markers in 20 meta-analyses of interventions is shown in Table 1. In the summary in Table 1, curcumin intake could significantly reduce major inflammatory markers.

In seven of ten meta-analyses, curcumin intake significantly lowered C-reactive protein (CRP) levels. In four of five meta-analyses, curcumin intake significantly lowered high-sensitivity C-reactive protein (hs-CRP) levels. In five of eight meta-analyses, curcumin intake significantly lowered interleukin 6 (IL-6) levels. In four of five meta-analyses, curcumin intake significantly increased adiponectin levels. In six of nine meta-analyses, curcumin intake significantly lowered tumor necrosis factor α (TNF-α) levels. In three of three meta-analyses, curcumin intake significantly lowered leptin levels.

A meta-analysis of RCTs by Dehzad et al. 2023 [42] showed the anti-inflammatory effect of curcumin/curcuminoids/turmeric. Subjects taking 60–2000 mg/day of curcumin or 150–1500 mg/day of curcuminoids or 900–3000 mg/day of turmeric showed reductions in CRP [weighted mean difference (WMD) = −0.58 mg/L; 95% confidence interval (CI) −0.74 to −0.41; I^2^ = 98.9%; *p* < 0.001], TNF-α (WMD = −3.48 pg/mL; 95% CI −4.38 to −2.58; I^2^ = 99.4%; *p* < 0.001) and IL-6 (WMD = −1.31 pg/mL; 95% CI −1.58 to −0.67; I^2^ = 88.2%, *p* < 0.001), with no difference in interleukin 1β (IL-1β) (WMD = −0.46 pg/mL; 95% CI −1.18 to 0.27; I^2^ = 75.8%; *p* < 0.001) in 46 publications [23,29,38,43,44,45,46,47,48,49,50,51,52,53,54,55,56,57,58,59,60,61,62,63,64,65,66,67,68,69,70,71,72,73,74,75,76,77,78,79,80,81,82,83,84,85] for CRP, 23 publications [24,43,47,48,50,54,55,56,60,63,64,67,68,71,79,83,86,87,88,89,90,91,92] for TNF-α, and 20 publications for IL-6 [23,24,45,46,47,48,50,54,63,64,68,83,86,89,91,93,94,95,96,97].

Ferguson et al. 2020 [116] also observed a significant anti-inflammatory effect of curcumin ranging from 46–4560 mg/day of curcumin with decreases in CRP in twenty-two publications [27,38,43,45,46,48,50,51,52,53,57,60,63,65,66,67,70,103,106,107,117,118] (WMD = −1.55 mg/L; 95% CI −1.81 to −1.30), IL-6 in eleven publications [24,45,46,50,63,86,103,107,114,117,118] (WMD = −1.69 pg/mL; 95% CI −2.56 to −0.82), TNF-α in twelve publications [24,43,46,50,63,67,86,112,114,118,119,120] (WMD = −3.13 pg/mL; 95% CI −4.62 to −1.64), IL-8 in four publications [46,63,86,107] (WMD = −0.54 pg/mL; 95% CI −0.82 to −0.28), and monocyte chemoattractant protein-1 (MCP-1) in three publications [24,63,86] (WMD = −2.48 pg/mL; 95% CI −3.96 to −1.00), and there was no change in intracellular adhesion molecule-1 in two publications [120,121].

According to a meta-analysis of seven publications [25,43,44,45,46,47,106] conducted by Gorabi et al. 2022 [105], curcumin decreased CRP levels (WMD = 3.67 mg/L; 95% CI −6.96 to −0.38; *p* = 0.02; I^2^ = 99.5%; *p* < 0.001) in comparison with a placebo. Moreover, sub-analyses showed that curcumin at a dose of ≤ 1000 mg/day significantly decreased CRP levels (WMD = −2.77 mg/L; 95% CI −4.34 to −1.19; *p* < 0.001) compared with curcumin dose > 1000 mg/day. The intervention duration of > 10 weeks showed significantly decreased CRP levels (WMD = −3.48 mg/L; 95% CI −5.64 to −1.33; *p* < 0.001) compared with the placebo. Levels of hs-CRP were significantly decreased after curcumin at a dose of ≤2000 mg/day in a meta-analysis of 23 studies (22 publications) [27,29,38,48,49,50,51,52,53,54,55,56,57,58,59,60,61,103,107,108,109,110]. Moreover, a sub-analysis showed a significant effect of curcumin on hs-CRP in RCTs with ≤1000 mg/day and those with a ≤10-week intervention period [27,29,38,48,49,50,51,52,53,54,55,56,57,58,59,60,61,103,107,108,109,110].

Gorabi et al. 2021 [113] observed a significant reduction in TNF-α in 12 publications [24,43,46,47,50,54,55,56,86,87,103,114] (WMD = −1.61 pg/mL; 95% CI −2.72 to −0.51; *p* < 0.001) after intake of curcumin (80–1500 mg/day), curcuminoids (300–1500 mg/day), or turmeric (112–1500 mg/day) for 4–12 weeks. Consistently, Sahebkar et al. 2016 [137] also observed that 300–1500 mg/day of taking curcumin for 4–12 weeks significantly decreased TNF-α (WMD = −4.69 pg/mL; 95% CI −7.10 to −2.28; *p* < 0.001) in a meta-analysis of nine publications [24,63,86,87,114,123,134,138,139].

On the other hand, Gorabi et al. 2021 [113] observed no differences in IL-6 in twelve publications [24,45,46,47,48,50,54,86,103,107,109,114] (WMD = −0.33 pg/mL; 95% CI −0.99 to −0.34; *p* = 0.33) after intake of curcumin (80–1000 mg/day), curcuminoids (300–1500 mg/day), or turmeric (112–1500 mg/day) for 4–12 weeks and IL-8 in four publications [46,86,87,107] (WMD = 0.52 pg/mL; 95% CI −1.13 to 2.17; *p* = 0.53) after intake of curcuminoids (1000 mg/day) or turmeric (112–1500 mg/day) for 4–8 weeks. Moreover, White et al. 2019 [122] found no effect of curcumin intake ranging from 20 to 2000 mg/day on CRP in five publications [25,45,46,53,106], hs-CRP in six publications [27,51,52,54,57,103], IL-1β in two publications [86,98], IL-6 in seven publications [24,45,86,93,103,114,123], and TNF-α in seven publications [46,86,87,103,114,123,124] compared with the controls with a follow-up duration ranging from 4 to 13 weeks.

A meta-analysis by Dehazd et al. 2023 [28] examined the effect of curcumin on adiposity-associated adipokines. Curcumin intake of 50–1500 mg/day for 4–42 weeks significantly increased serum adiponectin levels in 11 publications [29,30,31,32,34,36,37,38,39,40,41] (WMD = 2.48 μg/mL; 95% CI 1.34 to 3.62; *p* < 0.001; I^2^ = 96.3%, *p* < 0.001). Simental-Mendía et al. 2019 [128] also observed that curcumin intake of 500–1500 mg/day for 4–36 weeks significantly increased adiponectin levels (WMD = 6.47 ng/mL; 95% CI 1.85 to 11.10; *p* = 0.010; I^2^ = 94.85%) in a meta-analysis of five publications [31,39,41,129,130]. Moreover, a meta-analysis by Clark et al. 2019 [127] showed elevated adiponectin levels after curcumin intake of 200–1500 mg/day for 6–39 weeks compared with the placebo (WMD = 0.82 Hedges’ g; 95% CI 0.33 to 1.30; *p* ˂ 0.001) in five publications [30,31,36,38,41]. They found that RCTs for less than 10 weeks of the follow-up period showed greatly increased adiponectin levels (WMD = 1.05 Hedges’ g; 95% CI 0.64 to 1.45; *p* ˂ 0.001).

Dehzad et al. 2023 [28] showed that curcumin intake of 50–1500 mg/day for 4–42 weeks significantly decreased serum leptin levels in health subjects or subjects with metabolic syndrome, NAFLD, impaired glucose tolerance test (IGTT), prediabetes, T2DM, and major depressive disorder in nine publications [29,30,31,32,33,34,35,36,37] (WMD = −4.46 ng/mL; 95% CI −6.70 to −2.21; *p* < 0.001; I^2^ = 96.1%, *p* < 0.001). Consistent with Dehzad et al. 2023 [28], Atkin et al. 2017 [132] also observed a significant decrease in plasma leptin levels following a curcuminoid intake of 250–1000 mg/day for 4–24 weeks [standardized mean difference (SMD) = 0.69; 95% CI −1.16 to −0.23; *p* = 0.003; I^2^ = 76.53%)] in subjects with T2DM, metabolic syndrome, and major depressive disorder.

### 3.3. Antioxidant Effects

The influence of curcumin on antioxidant activity in six meta-analyses of interventions is shown in Table 2.

To summarize Table 2, curcumin appeared to have antioxidant effects. In five of six meta-analyses, curcumin intake significantly lowered malondialdehyde (MDA) levels. In two of three meta-analyses, curcumin intake significantly increased total antioxidant capacity (TAC) levels. In three of four meta-analyses, curcumin intake significantly increased superoxide dismutase (SOD) levels. One meta-analysis showed no effects on SOD levels. In two meta-analyses, one meta-analysis showed curcumin intake significantly increased glutathione peroxidase (GPx) levels, while the other meta-analysis showed neutral effects on GPx levels. One meta-analysis showed significantly increased catalase (CAT) levels. One meta-analysis showed no effects on glutathione reductase (GR) levels.

A very recent meta-analysis of RCTs by Dehzad et al. 2023 [42] showed the antioxidant effect of curcumin (80–3000 mg/day) or curcuminoids (180–1500 mg/day) or turmeric (1500–3000 mg/day) with a decrease in MDA [end-product of lipid peroxidation induced by reactive oxygen species (ROS)] (WMD = −0.33 µmol/L; 95% CI −0.53 to −0.12; I^2^ = 99.6%; *p* < 0.001) and increases in TAC (WMD = 0.21 mmol/L; 95% CI 0.08 to 0.33; I^2^ = 99.6%; *p* < 0.001) and SOD activity (WMD = 20.51 u/L; 95% CI 7.35 to 33.67; I^2^ = 95.4%; *p* < 0.001) in eighteen publications [23,43,53,57,60,68,76,77,78,79,80,81,82,92,98,143,144,146] for MDA, sixteen publications [23,30,43,49,57,68,76,77,78,79,80,82,88,143,144,145] for TAC, and seven publications [60,63,64,79,95,145,147] for SOD.

For the MDA levels, Tabrizi et al. 2019 [125] showed a significant reduction in MDA levels from the meta-analysis of five publications (SMD = −3.14; 95% CI −4.76 to −1.53; *p* < 0.001). Qin et al. 2018 [159] observed a significant reduction in MDA levels (SMD = −0.769; 95% CI −1.059 to −0.478) in seven publications [27,57,98,114,124,153,156]. Alizadeh et al. 2019 [150] showed a significant reduction in MDA (SMD = −0.46; 95% CI −0.68 to −0.25) in 12 publications [27,43,98,124,143,151,152,153,154,155,156,157].

For the TAC levels, Jakubczyk et al. 2020 [148] found that curcumin (mean 645 mg/day for 67 days) significantly elevated TAC in a meta-analysis of three publications [43,109,143] (SMD = 2.696; Z = 2.003; 95% CI = 0.058 to 5.335; *p* = 0.045).

For the SOD levels, Qin et al. 2018 [159] observed a significant increase in SOD activity (SMD = 1.084; 95% CI 0.487 to 1.680) in four publications [27,124,141,156]. Alizadeh et al. 2019 [150] also found a significant increase in SOD (an additional antioxidant enzyme) (SMD = 0.82; 95% CI 0.27 to 1.38) in seven publications [62,124,141,145,155,156,158]. Moreover, they found increased levels of CAT (SMD = 10.26; 95% CI 0.92 to 19.61) in five publications [62,141,143,145,155] and GPx (an antioxidant enzyme) in five publications [62,141,145,151,158] (SMD = 8.90; 95% CI 6.62 to 11.19) following curcumin intake compared with the control group [150].

On the other hand, several meta-analyses found no effects of curcumin on certain antioxidant biomarkers in MDA (80–1000 mg/day of curcumin intake for 10–12 weeks) [148], TAC (80–1500 mg/day of curcumin or 90–3000 mg/day of curcuminoid intake for 1–12 weeks) [150], GPx activity in red blood cells (RBCs) (46–66.3 mg/day of curcumin intake for 4–8 weeks) [159], SOD (1000 mg/day of curcuminoid intake for 4–8 weeks) [125], and GR (320 mg/day of curcumin intake for 8 weeks) [150].

### 3.4. Glucose Control

The influence of curcumin on glucose control in 18 meta-analyses of interventions is shown in Table 3.

To summarize Table 3, curcumin intake appeared to be effective in glucose control. In 14 of 15 meta-analyses, curcumin intake significantly reduced fasting blood glucose (FBG) levels. In 12 of 12 meta-analyses, curcumin intake significantly reduced homeostasis model assessment of insulin resistance (HOMA-IR). In seven of eight meta-analyses, curcumin intake significantly reduced glycated hemoglobin (HbA1c) levels. In eight of ten meta-analyses, curcumin intake significantly reduced insulin levels.

Sun et al. 2022 [100] extensively assessed RCTs for glucose control effects of curcumin in subjects with metabolic syndrome, obesity, prediabetes, T2DM, hyperlipidemia, hemodialysis, and diabetic foot ulcers (DFUs). They observed significant reductions in FBG (twenty-three publications [23,25,27,39,45,49,60,61,70,76,80,87,101,126,169,170,182,183,184,185,186,187,188]), Hb1Ac (sixteen publications [27,29,39,49,60,70,76,80,101,126,182,183,185,186,187,188]), HOMA-IR (eleven publications [39,45,49,60,70,76,80,101,183,188,189]), and insulin (five publications [45,76,80,187,188]) levels after curcumin intake of 80–2400 mg/day for 4–36 weeks.

Tian et al. 2022 [190] examined the effects of curcumin on glycemic control in a meta-analysis of RCTs with subjects with T2DM. The estimated pooled mean changes with curcumin −8.85 mg/dL (95% CI −14.4 to −3.29; *p* = 0.002) for FBG [38,49,57,87,101,114,182,183,191] and −0.54 (95% CI −0.81 to −0.27; *p* ≤ 0.001) for HbA1c (%) [38,49,57,87,101,114,182,183,191] were observed and compared with the controls. Zhang et al. 2021 [200] and Altobelli et al. 2021 [201] also conducted meta-analyses of RCTs in subjects with T2DM. Zhang et al. 2021 [200] found that curcumin intake of 80–1500 mg/day or curcuminoid intake of 300–1500 mg/day for 8–24 weeks significantly decreased HbA1c levels (WMD = −0.70; 95% CI −0.87 to −0.54; *p* < 0.0001) in 524 subjects with T2DM in five publications [31,60,70,183,185]. Interestingly, they found that HOMA-IR in the curcumin group was lower than that in the control group in the Middle East subgroup in one publication [60] (WMD = −0.60; 95% CI −0.74 to −0.46; *p* < 0.00001) and in the Asia subgroup in two publications [31,183] (WMD = −2.41; 95% CI −4.44 to −0.39; *p* = 0.02). The FBG in the curcumin group was lower than that in the control group in the Asia subgroup (SMD = −0.57; 95% CI −0.79 to −0.36; *p* < 0.00001), with no statistical significance in the Middle East subgroup in six publications [31,60,70,87,183,185]. Altobelli et al. 2021 [201] showed a significant reduction in HbA1c [effect size (ES) = −0.42; 95% CI −0.77 to −0.11; *p* = 0.008] in subjects with uncomplicated T2DM treated with curcumin in five publications [49,114,144,182,183]. In addition, HOMA-IR showed a statistically significant reduction in the curcumin-treated group (ES = −0.41; 95% CI −0.60 to −0.22; *p* < 0.001) [31,49,144,183].

Huang et al. 2019 [205] conducted a meta-analysis of subjects at risk for CVD. They found significant effects of curcumin treatment on FBG (SMD = −0.382; 95% CI −0.654 to −0.111 mg/dL, *p* = 0.006), HbA1c (SMD = −0.370; 95% CI −0.631% to −0.110%; *p* = 0.005) and HOMA-IR (SMD = −0.351; 95% CI −0.615 to −0.087; *p* = 0.009). Even though a meta-analysis by Tabrizi et al. 2018 [207] showed that curcumin intake (70–2000 mg/day of curcumin, 300–1000 mg/day of curcuminoids, 112–2400 mg/day of turmeric) for 4–24 weeks significantly reduced FBG levels (SMD = −0.78; 95% CI −1.20 to −0.37; *p* < 0.001) in nineteen publications [25,32,39,45,46,51,57,87,101,114,126,129,161,178,182,183,206,208,209], curcumin intake (294 mg/day of curcumin, 500–1000 mg/day of curcuminoids, 2000–2100 mg/day of turmeric) for 4–36 weeks was significantly associated increased insulin levels (SMD = 0.92; 95% CI 0.06 to 1.78; *p* = 0.036) in subjects with T2DM, obesity, hyperlipidemia, and NAFLD in seven publications [32,39,45,57,129,178,208]. Interestingly, Ashtary-Larky et al. 2021 [111] found that nano-curcumin intake of 40–120 mg/day for 6–12 weeks was associated with an improvement in FBG (WMD = −18.14 mg/dL; 95% CI −29.31 to −6.97; *p* = 0.001), insulin (WMD = −1.21 mg/dL; 95% CI −1.43 to −1.00; *p* < 0.001), and HOMA-IR (WMD = −0.28 mg/dL; 95% CI −0.33 to −0.23; *p* < 0.001).

### 3.5. Lipid Profiles

The influence of curcumin on lipid profiles in 21 meta-analyses of interventions is shown in Table 4.

To summarize Table 4, curcumin intake appeared to exert a total cholesterol (TC)-lowering effect, while curcumin intake has neutral effects on other lipids, including TG, low-density lipoprotein cholesterol (LDL-C) and high-density lipoprotein cholesterol (HDL-C). In 14 of 19 meta-analyses, curcumin intake significantly reduced TC levels, while five meta-analyses showed no effects on TC levels. In nine of twenty meta-analyses, curcumin intake significantly reduced triglyceride (TG) levels, while eleven meta-analyses showed no effects on TG levels. In four of eighteen meta-analyses, curcumin intake significantly decreased LDL-C levels, while fourteen meta-analyses showed no effects on LDL-C levels. Only three of twenty meta-analyses showed significantly increased HDL-C levels.

Tabrizi et al. 2018 [207] extensively included RCTs (19 publications [25,31,45,51,52,57,87,101,114,126,138,161,178,183,206,208,214,219,220] for TG, 20 publications [25,45,51,52,57,87,101,114,126,138,161,178,182,183,206,208,209,214,219,220] for TC and HDL-C, and 21 publications [25,45,46,51,52,57,87,101,114,126,138,161,178,182,183,206,208,209,214,219,220] for LDL-C) in the meta-analysis. Curcumin intake (70–2000 mg/day of curcumin, 300–1000 mg/day of curcuminoids, 112–2100 mg/day of turmeric) for 4–24 weeks significantly decreased levels of TG (SMD = −1.21; 95% CI −1.78 to −0.65; *p* < 0.001) and TC (SMD = −0.73; 95% CI −1.32 to −0.13; *p* = 0.01) in subjects with obesity, metabolic syndrome, hyperlipidemia, NAFLD, T2DM, acute coronary syndrome, coronary artery disease (CAD), and diabetic nephropathy. However, they found no significant effect of curcumin on LDL-C and HDL-C.

Three meta-analyses [190,200,201] of RCTs were performed to examine the effects of curcumin on lipids in subjects with T2DM. A meta-analysis of nine publications (*n* = 604) [38,49,57,87,101,114,182,183,191] by Tian et al. 2022 [190] showed significant reductions in TG (WMD = −18.97 mg/dL; 95% CI −36.47 to −1.47; *p* = 0.03) and TC (WMD = −8.91 mg/dL; 95% CI −14.18 to −3.63, *p* = 0.001), while no differences in LDL-C and HDL-C were found compared with the controls after the intake of curcumin (80–1500 mg/day), curcuminoids (300–1000 mg/day), or turmeric (1500–2100 mg/day) for 4–12 weeks. Zhang et al. 2021 [200] reported that TC and TG levels of the curcumin group with T2DM, who consumed a curcumin dose of 1500 mg/day or a curcuminoid dose of 300 mg/day for 12–24 weeks decreased in the Asia subgroup [31,183] (TC: WMD = −23.45; 95% CI −40.04 to −6.84; *p* = 0.006; TG: WMD = −54.14; 95% CI −95.71 to −12.57; *p* = 0.01), while there was no statistical significance in the Middle East subgroup [87,101] after the intake of 1500 mg/day of curcumin or 300 mg/day of curcuminoids for 8–12 weeks. Curcumin supplementation (1500 mg/day of curcumin or, 300 mg/day of curcuminoids) for 12–24 weeks also reduced LDL-C in subjects with T2DM in the Asia subgroup [31,183] (WMD = −20.85; 95% CI −28.78 to −12.92; *p* < 0.00001). The difference in HDL-C was not significant between groups by curcumin supplementation (1500 mg/day of curcumin, 300–1000 mg/day of curcuminoids, 1500 mg/day of turmeric) for 8–24 weeks. Altobelli et al. 2021 [201] showed significant reductions in TG (ES = −0.57; 95% CI −0.83 to −0.31; *p* < 0.001), TC (ES = −0.30; 95% CI −0.53 to −0.07; *p* < 0.001), and LDL-C (ES = −0.28; 95% CI −0.52 to −0.04; *p* = 0.021), but there was no effect on HDL-C in subjects with uncomplicated T2DM treated with curcumin (80–1500 mg/day of curcumin, 300 mg/day of curcuminoids, 2100 mg/day of turmeric) for 8–12 weeks compared with the placebo in a meta-analysis of five publications [31,38,49,114,183].

The effects of curcumin on lipid profiles in subjects with chronic kidney diseases were meta-analyzed by Futuhi et al. 2022 [102]. Curcumin intake (80–500 mg/day of curcumin, 2500 mg/day of turmeric) for 8–12 weeks significantly reduced TC (WMD = −13.77 mg/dL; 95% CI −26.77 to −0.77; *p* = 0.04) in five publications [71,72,76,151,212] compared with the controls. However, the significant effect of curcumin intake (80–500 mg/day of curcumin, 2500 mg/day of turmeric) for 8–12 weeks was not confirmed on TG, LDL-C, and HDL-C.

Qin et al. 2017 [221] performed a meta-analysis to investigate the effects of curcumin on lipids in subjects with cardiovascular risk. In seven publications [25,31,57,114,126,161,182], curcumin intake (80 mg/day of curcumin, 70–1890 mg/day of curcuminoids, 2000–2400 mg/day of turmeric) for 4–12 weeks significantly reduced LDL-C (SMD = −0.340; 95% CI −0.530 to −0.150; *p* < 0.0001) and TG (SMD = −0.214; 95% CI −0.369 to −0.059; *p* = 0.007) compared with the controls. However, HDL-C and TC levels were not significantly improved after curcumin intake (80 mg/day of curcumin, 70–1890 mg/day of curcuminoids, 2000–2400 mg/day of turmeric) for 4–12 weeks. Interestingly, Ashtary-Larky et al. 2021 [111] observed that nano-curcumin intake of 40–120 mg/day for 6–12 weeks elevated HDL-C (WMD = 5.77 mg/dL; 95% CI 2.90 to 8.64; *p* < 0.001) in six publications [23,50,71,76,171,182] in 337 subjects with metabolic syndrome, hemodialysis, NAFLD, and T2DM.

On the other hand, several meta-analyses showed no effect on lipids. Saeedi et al. 2022 [211] updated the protective effects of curcumin on blood lipid levels, including 10 publications [25,51,101,126,164,165,177,184,191,209] after curcumin intake (50–2000 mg/day of curcumin, 1000 mg/day of curcuminoids, 2100 mg/day of turmeric) for 4–12 weeks. However, they did not find significant effects on TG, TC, LDL-C, and HDL-C. Sahebkar et al. 2014 [222] also found no effect of curcumin (45–6000 mg/day of curcuminoids) for 1–24 weeks on TC, LDL-C, TG, and HDL-C in five publications [52,114,214,215,218].

### 3.6. Body Weight, Body Mass Index, and Waist Circumference

The influence of curcumin on BW, body mass index (BMI), and waist circumference (WC) in meta-analyses of interventions is shown in Table 5.

To summarize Table 5, curcumin intake appeared to have neutral effects on BW, BMI, and WC. In four of ten meta-analyses, curcumin intake significantly reduced BW levels, while six meta-analyses showed no effect on BW levels. In five of twelve meta-analyses, curcumin intake significantly reduced WC levels, while seven meta-analyses showed no effect on WC levels. In seven of fourteen meta-analyses, curcumin intake significantly reduced BMI levels, while seven meta-analyses showed no effect on BMI levels.

Sun et al. 2022 [100] found decreased BW (WMD = −0.94 kg; 95% CI −1.40 to −0.47) and BMI (WMD = −0.40 kg/m^2^; 95% CI −0.60 to −0.19) after 80–2400 mg/day of curcumin for 4–24 weeks in subjects with metabolic syndrome, obesity, hyperlipidemia, prediabetes, T2DM, diabetic sensorimotor polyneuropathy (DSPN), and hemodialysis in 15 publications [23,25,38,49,60,61,70,101,126,170,182,183,185,209,223] for BW and 16 publications [23,25,29,49,60,61,76,101,126,170,182,183,185,188,209,223] for BMI. However, no effect on WC was observed after a curcumin intake of 80–2400 mg/day for 4–17 weeks. Similar findings were observed in a meta-analysis by Mousavi et al. 2020 [225] in overweight and obese subjects with metabolic syndrome, obesity, prediabetes, NAFLD, and T2DM who consumed curcumin (200–1000 mg/day of curcumin, 1000–1500 mg/day of curcuminoids) for 4–39 weeks. Azhdari et al. 2019 [204] also showed no significant WC change in subjects with metabolic syndrome who consumed curcumin at a dose of 800–2400 mg/day for 4–8 weeks in four publications [25,61,172,223].

### 3.7. Blood Pressure and Endothelial Function

The influence of curcumin on BP and endothelial function in meta-analyses of interventions is shown in Table 6.

To summarize Table 6, curcumin intake appeared to have neutral effects on BP. In two of two meta-analyses, curcumin intake significantly increased FMD levels. Even though both meta-analyses showed a positive effect of curcumin intake on FMD, meta-analyses with more RCTs will be needed to determine the effect of curcumin on FMD. In two of five meta-analyses, curcumin intake significantly lowered diastolic blood pressure (DBP) levels. In one of five meta-analyses, curcumin intake significantly lowered systolic blood pressure (SBP) levels.

In a meta-analysis of three publications [25,27,61] by Azhdari et al. 2019 [204], a significant change in DBP (WMD = −2.96 mmHg; 95% CI −5.09 to −0.83; *p* = 0.007) in subjects with metabolic syndrome who consumed curcumin (1000 mg/day of curcumin, 1000 mg/day of curcuminoids, 2400 mg/day of turmeric) for 8 weeks was noted, while no change in SBP was observed.

Ashtary-Larky et al. 2021 [111] observed that a nano-curcumin dose of 40–80 mg/day for 6–12 weeks significantly reduced SBP (WMD = −7.09 mmHg; 95% CI −12.98 to −1.20; *p* < 0.001) in four publications [23,50,76,171]. However, a nano-curcumin dose of 40–80 mg/day for 12 weeks showed no effect on DBP in three publications [50,76,171].

On the other hand, Hadi et al. 2019 [242] extensively included 11 publications [25,50,61,65,66,89,151,178,219,243,244] on subjects with metabolic syndrome, obesity, T2DM, healthy elderly, NAFLD, COPD, diabetic proteinuria, chronic kidney disease (CKD), and lupus nephritis. They failed to find significant reductions in SBP and DBP after curcumin intake (80–1000 mg/day of curcumin, 320–2400 mg/day of turmeric) for 8–24 weeks.

### 3.8. Depression and Cognitive Function

The influence of curcumin on depression and cognitive function in meta-analyses of interventions is shown in Table 7.

To summarize Table 7, curcumin intake appeared to have positive effects on depression. In four of four meta-analyses, curcumin intake significantly improved depression. In one of two meta-analyses, curcumin intake significantly improved cognitive function. In one of one meta-analysis, curcumin intake significantly improved response rates. In one of one meta-analysis, curcumin intake significantly lowered anxiety symptoms. In one of one meta-analysis, curcumin intake significantly improved working memory. In one of one meta-analysis, curcumin intake significantly improved working memory.

Four meta-analyses [245,255,257,258] of RCTs showed significantly improved depression. Wang et al. 2021 [245] found that curcumin intake of 80–3000 mg/day for 4–24 weeks improved depressive symptoms (SMD = −0.32; 95% CI −0.50 to −0.13; I^2^ = 15%, *p* = 0.30) in four publications [251,252,253,254] and response rates [odds ratio (OR) = 3.20; 95% CI 1.28 to 7.99; I^2^ = 35%, *p* = 0.21] in three publications [139,248,250]. Fusar-Poli et al. 2020 [255] showed an improvement in depression after curcumin intake of 150–1500 mg/day for 4–12 weeks in nine publications [139,246,247,248,249,250,252,254,256] (Hedge’s g = −0.75; 95% CI −1.11 to −0.39; *p* < 0.001; I^2^ = 26.28%) and anxiety symptoms after curcumin intake of 500–1000 mg/day for 4–12 weeks in four publications [248,249,252,256] (Hedge’s g = −2.62; 95% CI −4.06 to −1.17; *p* < 0.001) with large ES. A meta-analysis of six publications [139,246,249,250,252,256] by Ng et al. 2017 [257] showed an improved effect of curcumin intake of 500–1000 mg/day for 4–8 weeks on depressive symptoms compared with a placebo or control (pooled SMD = −0.344; 95% CI −0.558 to −0.129; *p* = 0.002). This improved effect was based on the Hamilton Rating Scale for Depression (HAM-D) score from the baseline after curcumin intake. A meta-analysis of six RCTs [139,246,249,250,252,256] by Al-Karawi et al. 2017 [258] showed decreased major depression (SMD = −0.34; 95% CI −0.56 to −0.13; *p* = 0.002) after curcumin intake of 500–1000 mg/day for 4–8 weeks. A bigger effect was seen when a longer and higher dose of curcumin intake in middle-aged subjects was supplemented.

With regard to the effect of curcumin on cognitive function, Tsai et al. 2021 [259] examined the effect of curcumin on cognition in a meta-analysis of RCTs. Working memory in three publications [261,266,267] was enhanced after curcumin intake of 80–180 mg/day for 8–16 weeks compared with the placebo (Hedges’ g = 0.396; 95% CI 0.078 to 0.714; *p* = 0.015; I^2^ = 0.0%). Zhu et al. 2019 [268] suggested that curcumin intake of 80–1320 mg/day for 4–72 weeks appeared to improve cognitive function in older subjects rather than in subjects with AD or schizophrenia. Curcumin was more effective in improving cognitive function in the elderly (SMD = 0.33; 95% CI 0.05 to 0.62; *p* = 0.02) in three publications [118,263,269]. Cognitive state measures calculated using the Mini-Mental State Examination (MMSE) were worse in subjects with AD taking curcumin than in placebo subjects (SMD = −0.90; 95% CI −1.48 to −0.32; *p* = 0.002) in two publications [264,265]. In a meta-analysis of four RCTs [172,262,271,272] by Sarraf et al. 2019 [270], curcumin intake (200–1820 mg/day during 8–12 weeks) was associated with higher serum brain-derived neurotrophic factor (BDNF) concentrations (WMD = 1789.38 pg/mL; 95% CI 722.04 to 2856.71; *p* < 0.01; I^2^ = 83.5%; *p* < 0.001).

## 4. Discussion

The present study aimed to examine the effects of curcumin on chronic inflammatory metabolic disease by extensively evaluating meta-analyses of RCTs. Fifty-four meta-analyses were included in this review, which investigated the effects of curcumin supplementation on inflammation, antioxidants, glucose control, lipids, anthropometric parameters, BP, endothelial function, depression, and cognitive function.

This review clearly observed that curcumin intake could significantly reduce major inflammatory markers of CRP (seven of ten meta-analyses of RCTs), IL-6 (five of eight meta-analyses of RCTs), and TNF-α (six of nine meta-analyses of RCTs). In light of this review finding that curcumin intake significantly lowered MDA levels in five of six meta-analyses, curcumin intake appeared to have antioxidant activity.

Consistent with our findings, Naghsh et al. 2023 [273] found the effects of curcumin on inflammatory biomarkers in an umbrella meta-analysis of 10 RCTs [111,113,115,116,122,125,133,137,140,274], in which curcumin doses ranging from 100 to 1900 mg/day were supplemented for 4.5 to 10.5 weeks. Curcumin supplementation decreased levels in CRP (corresponding ES = −0.74; 95% CI −1.11 to −0.37; *p* < 0.001; I^2^ = 62.1%, *p* = 0.015) in seven meta-analyses [111,115,116,122,125,140,274], IL-6 (ES = −1.07; 95% CI −1.71 to −0.44; *p* < 0.001; I^2^ = 75.6%, *p* < 0.001) in six meta-analyses [111,113,116,122,125,133], and TNF-α (ES = −1.92; 95% CI −2.64 to −1.19; *p* < 0.01; I^2^ = 18.1%; *p* = 0.296) in six meta-analyses [111,113,116,122,125,137]. In particular, greater reductions in CRP and TNF-α were observed in subjects aged over 45 years compared with younger subjects [273].

Several studies have elucidated the mechanisms responsible for the beneficial effects of curcumin on inflammation. Curcumin inhibited inflammatory eicosanoid-generating enzymes, such as cyclooxygenase and lipoxygenase [275]. Curcumin suppressed the nitric oxide (NO) production and expression of inducible nitric oxide synthase (iNOS) protein and messenger ribonucleic acid (mRNA) [276,277]. Curcumin decreased the activation of nuclear factor-κB (NF-κB), a transcription factor that regulates the expression of inflammatory response-related genes resulting in the down-regulation of pro-inflammatory cytokines, such as IL-6 and TNF-α [278]. Furthermore, curcumin has an antioxidant effect through free-radical scavenging activity and antioxidant enzyme regulatory activity [279,280]. Curcumin treatment resulted in significant increases in glutathione (GSH) concentration and GPx and SOD activity and a significant decrease in lipid peroxidation in mouse skin exposed to fractionated γ-radiation [281]. Due to the relationship between oxidative stress and inflammation through activating transcription factors associated with inflammation, the antioxidant effect of curcumin may contribute to the anti-inflammatory activity of curcumin.

In the present review, curcumin intake showed a glucose-lowering effect in most meta-analyses of RCTs. Curcumin intake significantly reduced FBG levels in 14 of 15 meta-analyses and significantly reduced HOMA-IR in 12 of 12 meta-analyses. In addition, curcumin intake significantly reduced HbA1c levels in seven of eight meta-analyses and significantly reduced insulin levels in eight of ten meta-analyses. In bisphenol A (BPA)-induced insulin resistant HepG2 cells, curcumin treatment improved glucose consumption and insulin signaling through the suppression of JNK/p38 pathways [282]. Moreover, significant reductions in the TNF-α, IL-6, MDA, and COX-2 levels confirmed that inflammation and oxidative stress have been associated with insulin resistance [282].

In the present review, we observed the TC-lowering effect of curcumin intake in 14 of 19 meta-analyses of RCTs, but we could not find the other lipids, such as TG, LDL-C, and HDL-C. Similarly, an umbrella meta-analysis by Musazadeh et al. 2022 [283] found the lipid-lowering effect of curcumin intake in adults, with significant improvement in TC (ES = −0.81 mg/dL; 95% CI −1.39 to −0.24; *p* = 0.006), TG (ES = −0.84 mg/dL; 95% CI −1.42 to −0.27; *p* = 0.004), LDL-C (ES = −0.49 mg/dL; 95% CI −0.85 to −0.13; *p* = 0.007), and HDL-C levels (ES = 1.34 mg/dL; 95% CI 0.37 to 2.31; *p* = 0.007).

This review could not find significant effects of curcumin intake on BW, BMI, and WC. Unlike our findings, curcumin intake significantly reduced WC (MD = −1.32 cm; 95% CI −1.95 to −0.69 cm), BMI (MD = −0.24 kg/m^2^; 95% CI −0.32 to −0.16 kg/m^2^), and BW (MD = −0.59 kg; 95% CI −0.81 to −0.36 kg) in an umbrella review and updated meta-analysis of 50 RCTs conducted by Unhapipatpong et al. 2023 [284] with the purpose of finding the effects of curcumin on weight loss. This conflicting result could be explained by factors affecting the efficacy of curcumin, which have been described in several studies. The beneficial effects of curcumin on BMI reduction were only shown in subjects with BMI ≥ 25 kg/m^2^ in Unhapipatpong et al. 2023 [284]. A dose–response meta-analysis concluded that curcumin can significantly reduce BW (WMD: −1.14 kg, 95% CI: −2.16, −0.12, *p* = 0.02) and BMI (WMD: −0.48 kg/m^2^, 95% CI: −0.78, −0.17, *p* = 0.002), but WC can only be reduced in subgroup of a dose of ≥1000 mg/day of curcumin for ≥8 weeks of duration in overweight subjects [225]. Additionally, the effects of curcumin on anthropometric indices differed, depending on the forms of curcumin, and they showed more efficacy in the bioavailable form [284,285].

The strength of this study is that it includes a meta-analysis of as many RCTs as possible to extensively investigate the effects of curcumin intake on chronic inflammatory metabolic diseases, including inflammatory markers, antioxidant effects, glucose control, lipid profiles, BW, BMI, WC, BP, endothelial function, depression, and cognitive function. This present study comprehensively reviewed and compared the results of individual meta-analyses of RCTs. This study provides evidence on the question of whether curcumin intake is really effective for chronic inflammatory metabolic disease by examining the meta-analyses of the contributors to risks for chronic inflammatory disease. The present study has limitations. We included articles in the English language in PubMed^®^/MEDLINE to identify and select relevant meta-analyses of RCTs. Several RCTs used in the meta-analyses included in this study were of low quality, which might result in limitations in generalizing the findings of this review. In order to generalize the findings, RCTs with robust designs should be implemented, and a meta-analysis should be performed based on these RCTs.

## 5. Conclusions

In conclusion, we evaluated the effects of curcumin intake on chronic inflammatory metabolic disease by reviewing meta-analyses of RCTs. Curcumin intake (40–6000 mg/day of curcumin, 150–1500 mg/day of curcuminoids, 900–3000 mg/day of turmeric) for 12 days–24 weeks significantly reduced CRP in seven of ten meta-analyses of RCTs. Curcumin intake (40–6000 mg/day of curcumin, 100–1500 mg/day of curcuminoids, 112–3000 mg/day of turmeric) for 12 days–32 weeks significantly reduced IL-6 in five of eight meta-analyses of RCTs. Curcumin intake (80–1500 mg/day of curcumin, 150–1500 mg/day of curcuminoids, 112–3000 mg/day of turmeric) for 4–12 weeks significantly reduced TNF-α in six of nine meta-analyses of RCTs. Curcumin intake (50–1500 mg/day of curcumin, 1500 mg/day of curcuminoids) for 4–42 weeks significantly increased adiponectin levels in four of five meta-analyses. Curcumin intake (46–1500 mg/day of curcumin, 90–4000 mg/day of curcuminoids, 1500–3000 mg/day of turmeric) for 3 days–16 weeks significantly decreased MDA levels in five of six meta-analyses. Curcumin intake (50–2400 mg/day of curcumin, 294–2000 mg/day of curcuminoids, 112–3000 mg/day of turmeric) for 4–36 weeks significantly reduced FBG levels in 14 of 15 meta-analyses. Curcumin intake (40–2100 mg/day of curcumin, 70–1500 mg/day of curcuminoids, 300–3000 mg/day of turmeric) for 4–36 weeks significantly reduced HOMA-IR in 12 of 12 meta-analyses. Curcumin intake significantly reduced TC levels in 14 of 19 meta-analyses. Our results reveal that curcumin intake effectively and protectively exerts chronic inflammatory metabolic diseases through improved levels of glucose homeostasis, MDA, TC, and inflammation (CRP, IL-6, TNF-α, and adiponectin). The safety and efficacy of curcumin indicate that it is possible to prevent and treat chronic inflammatory metabolic diseases.

## Figures and Tables

**Figure 1 nutrients-16-01728-f001:**
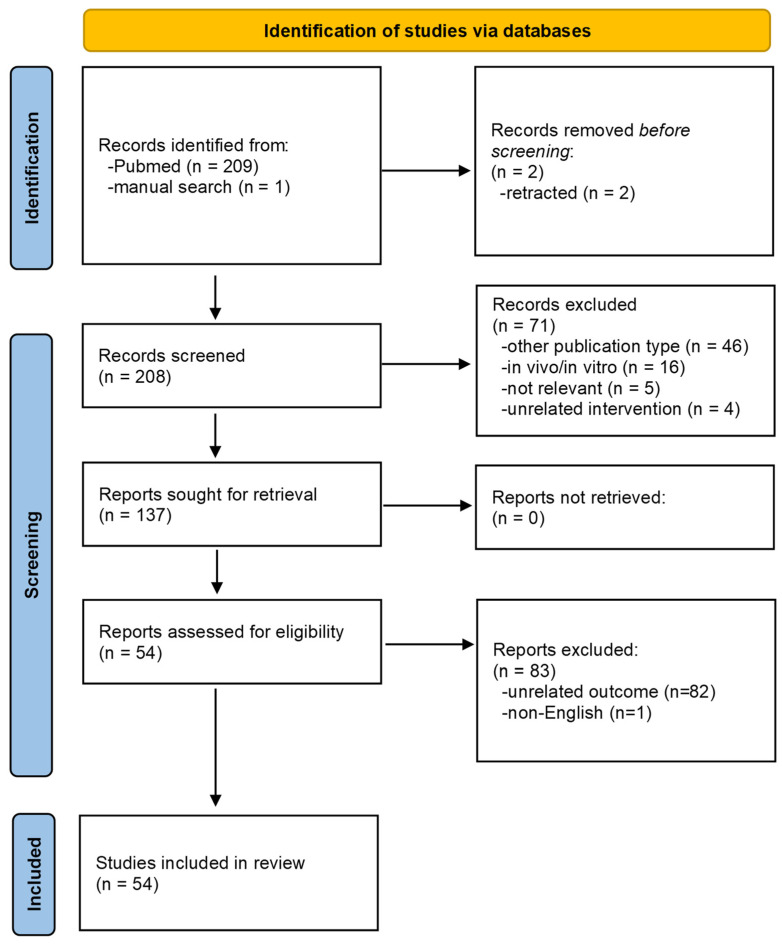
Flow diagram of the literature review.

**Table 1 nutrients-16-01728-t001:** Meta-analyses of RCTs investigating the effect of curcumin intake on inflammation.

Ref.	No. of Studies Included	Health Status of Subjects	No. of Subjects	Age of Subjects (Years)	Design	Period (Weeks)	Dose	Outcomes (Effect Size)	Quality of Primary Studies	Databases
Qiu et al. 2023 [22]	2 publications [23,24]	Mets	122	34–64	P	6–8	80 mg/day of curcumin, 1000 mg/day of curcuminoids	↔ IL-6 (MD = −1.5 pg/mL; 95% CI −3.97 to 0.97; *p* = 0.23) ↓ TNF-α (MD = −12.97; 95% CI −18.37 to −7.57; *p* < 0.00001)	Cochrane risk of bias tool (71.4%)	PubMed, SCOPUS, Cochrane Library, EMBASE, Web of Science, and China Biological Medicine
	2 publications [23,25]	Mets	130	54–64	P	6–8	80 mg/day of nano-curcumin, 2400 mg/day of turmeric	↓ CRP (MD = −1.24 mg/L; 95% CI −1.71 to −0.77; *p* < 0.00001)		
	2 publications [26,27]	Mets	143	34–61	P	8–12	80 mg/day of nano-curcumin, 1000 mg/day of curcuminoids	↔ hs-CRP (MD = −1.10 mg/L; 95% CI −4.35 to 2.16; *p* = 0.51)		
Dehzad et al. 2023 [28]	9 publications [29,30,31,32,33,34,35,36,37]	Healthy, Mets, NAFLD, IGTT, prediabetes, T2DM, major depressive disorder	1172	18–70	P	4–42	50–1500 mg/day of curcumin	↓ leptin (WMD = −4.46 ng/mL; 95% CI −6.70 to −2.21; *p* < 0.001; I^2^ = 96.1%; *p* < 0.001)	Cochrane risk of bias tool (61.0%)	PubMed, Scopus, Web of Science, Cochrane Library, and Google Scholar
	11 publications [29,30,31,32,34,36,37,38,39,40,41]	Healthy, Mets, IGTT, T2DM, prediabetic, NAFLD	783	18–70	P	4–42	50–1500 mg/day of curcumin	↑ adiponectin (WMD = 2.48 μg/mL; 95% CI 1.34 to 3.62; *p* < 0.001; I^2^ = 96.3%; *p* < 0.001)		
Dehzad et al. 2023 [42]	46 publications [23,29,38,43,44,45,46,47,48,49,50,51,52,53,54,55,56,57,58,59,60,61,62,63,64,65,66,67,68,69,70,71,72,73,74,75,76,77,78,79,80,81,82,83,84,85]	Mets, healthy, obese, hemodialysis, chronic pulmonary complications, solid tumor, OA, T2DM, CAD, COPD, hyperlipidemia, migraine, infertile men, NAFLD, coronary elective angioplasty, RA, PCOS, Takayasu arteritis, pre-HTN, UC, diabetic foot ulcer	2873	18–70	P, CO	4–24	80–2000 mg/day of curcumin, 150–1500 mg/day of curcuminoids, 900–3000 mg/day of turmeric	↓ CRP (WMD = −0.58 mg/L; 95% CI −0.74 to −0.41; *p* < 0.05)	Cochrane risk of bias tool (84.2%)	PubMed, Scopus, Web of Science, Cochrane Library, and Google Scholar
	23 publications [24,43,47,48,50,54,55,56,60,63,64,67,68,71,79,83,86,87,88,89,90,91,92]	Mets, T2DM, OA, PUD, UC, obese, hemodialysis, NAFLD, HIV, solid tumor, migraine, lupus erythematosus, Takayasu arteritis, infertile coronary elective angioplasty	1488	18–70	P, CO	4–12	80–1500 mg/day of curcumin, 150–1000 mg/day of curcuminoids, 1500–3000 mg/day of turmeric	↓ TNF-α (WMD = −3.48 pg/mL; 95% CI −4.38 to −2.58; *p* < 0.05)		
	20 publications [23,24,45,46,47,48,50,54,63,64,68,83,86,89,91,93,94,95,96,97]	Mets, obese, chronic pulmonary complications, solid tumor, OA, hemodialysis, hyperlipidemia, lupus erythematosus, migraine, pre-HTN, ankylosing spondylitis, chronic branchial asthma, chronic knee pain	1166	18–70	P, CO	4–12	60–1500 mg/day of curcumin, 100–1500 mg/day of curcuminoids, 2800–3000 mg/day of turmeric	↓ IL-6 (WMD = −1.31 pg/mL; 95% CI −1.58 to −0.67; *p* < 0.05)		
	6 publications [47,79,86,88,91,98]	Obese, knee OA, PUD, pre-HTN, coronary elective angioplasty	482	18–70	P, CO	4–16	80–1500 mg/day of curcumin	↔ IL-1β (WMD = −0.46 pg/mL; 95% CI −1.18 to 0.27; *p* > 0.05)		
Shen et al. 2022 [99]	2 publications [58,73]	PCOS	81	24–35	P	6–8	93.34–1000 mg/day of curcumin	↓ CRP (WMD = −0.785; 95% CI −1.553 to −0.017; *p* = 0.045)	Cochrane risk of bias tool (73.5%)	PubMed, Embase, Cochrane Library, Web of Science, Scopus, Clinical Trials, Chinese Clinical Trial Registry, Chinese Biomedical Literature Database, Chinese National knowledge Infrastructure, VIP database, and Wanfang Database
Sun et al. 2022 [100]	9 publications [27,29,38,49,60,61,76,80,101]	Mets, T2DM, DFU, CHD, hyperlipidemia, IGT	697	38–70	P, CO	6–24	80–2100 mg/day of curcumin	↓ hs-CRP (WMD = −1.11; 95% CI −2.16 to −0.05; *p* = 0.039)	Cochrane risk of bias tool (69.3%) Jadad scale (77.0%)	PubMed, Embase, Web of Science, CNKI, Wanfang, and CBM
	4 publications [29,31,38,39]	T2DM, IGT	525	57–70	P, CO	10–36	180–1500 mg/day of curcumin	↔ adiponectin (WMD = 6.43 ng/mL; 95% CI −1.19 to 14.05; *p* = 0.98)		
Futuhi et al. 2022 [102]	5 publications [71,72,81,103,104]	ESRD hemodialysis, CKD	212	32–73	P	8–12	120 mg/day of curcumin, 1000 mg/day of curcuminoids, 2500 mg/day of turmeric	↔ CRP (WMD = −0.13 mg/L; 95% CI −3.25 to 3.30; *p* = 0.93)	Cochrane risk of bias tool (73.0%)	PubMed, Scopus, Cochrane Library, and Web of Science
	3 publications [71,103,104]	ESRD, hemodialysis, CKD	141	32–73	P	8–12	120 mg/day of curcumin, 1500 mg/day of turmeric	↓ TNF-α (WMD = −18.87 pg/mL; 95% CI −28.36 to −9.38; *p* < 0.001)		
Gorabi et al. 2022 [105]	7 publications [25,43,44,45,46,47,106]	Mets, RA, overweight, infertility	493	18–65	P	4–12	80–1000 mg/day of curcumin, 250–500 mg/day of curcuminoids 900–2400 mg/day of turmeric.	↓ CRP (WMD = 3.67 mg/L; 95% CI −6.96 to −0.38; *p* = 0.02; I^2^ = 99.5%; *p* < 0.001) ↓ CRP (WMD = −2.77 mg/L; 95% CI −4.34 to −1.19; *p* < 0.001) at ≤1000 mg/day of curcumin compared with dose > 1000 mg/day of curcumin ↓ CRP (WMD = −3.48 mg/L; 95% CI −5.64 to −1.33; *p* < 0.001) with intervention duration of >10 weeks	Jadad scale (81.3%)	MEDLINE/ PubMed, Cochrane Library, Web of Science, and Scopus
	22 publications [27,29,38,48,49,50,51,52,53,54,55,56,57,58,59,60,61,103,107,108,109,110]	Mets, CAD, chronic cutaneous, ESRD, knee OA, T2DM, migraine	1142	18–73	P	3 days–8 weeks	1000–4000 mg/day of curcuminoids, 1500 mg/day of turmeric	↓ hs-CRP in RCTs with ≤1000 mg/day and those with ≤10-week intervention period		
Ashtary-Larky et al. 2021 [111]	5 publications [23,48,50,71,76]	NAFLD, Mets, migraine, T2DM, hemodialysis	279	28–68	P	6–12	40–120 mg/day of nano-curcumin	↓ CRP (WMD = −1.29 mg/L; 95% CI −2.15 to −0.44; *p* = 0.003)	Cochrane risk of bias tool (55.6%)	PubMed, Scopus, Embase, and Web of Science
	3 publications [23,48,50]	NAFLD, Mets, migraine	172	28–63	P	6–12	40–80 mg/day of nano-curcumin	↓ IL-6 (WMD = −2.78 mg/dL; 95% CI −3.76 to −1.79; *p* <0.001)		
	2 publications [50,112]	NAFLD, migraine	122	28–49	P	8–12	40–80 mg/day of nano-curcumin	↔ TNF-α (WMD = −3.09 mg/dL; 95% CI −8.75 to 2.57; *p* = 0.284)		
Gorabi et al. 2021 [113]	3 publications [47,86,98]	Knee OA, overweight	307	18–65	P	4–16	1000 mg/day of curcuminoids, 500–900 mg/day of turmeric	↓ IL-1 (WMD = −2.33 pg/mL; 95% CI −3.33 to −1.34; *p* < 0.001)	Jadad scale (not indicated)	PubMed/MEDLINE, Web of Science, Scopus, and Cochrane Library
	12 publications [24,43,46,47,50,54,55,56,86,87,103,114]	Mets, T2DM, overweight, knee OA, NAFLD, ESRD, diabetic nephropathy, infertility, UC	760	18–65	P	4–12	80–1500 mg/day of curcumin, 300–1500 mg/day of curcuminoids, 112–1500 mg/day of turmeric	↓ TNF-α (WMD = −1.61 pg/mL; 95% CI −2.72 to −0.51; *p* < 0.001)		
	12 publications [24,45,46,47,48,50,54,86,103,107,109,114]	Mets, T2DM, overweight, knee OA, hyperlipidemia, ESRD, NAFLD, migraine	808	18–65	P	4–12	80–1000 mg/day of curcumin, 300–1500 mg/day of curcuminoids, 112–1500 mg/day of turmeric	↔ IL-6 (WMD = −0.33 pg/mL; 95% CI −0.99 to −0.34; *p* = 0.33)		
	4 publications [46,86,87,107]	Diabetic nephropathy, chronic cutaneous, overweight	240	18–65	P	4–8	1000 mg/day of curcuminoids, 112–1500 mg/day of turmeric	↔ IL-8 (WMD = 0.52 pg/mL; 95% CI −1.13 to −2.17; *p* = 0.53)		
Abdelazeem et al. 2021 [115]	2 publications [58,73]	PCOS	81	24–35	P	6–8	93.34–1000 mg/day of curcumin	↔ CRP (MD = −0.55; 95% CI −1.64 to 0.54; *p* = 0.33)	Cochrane risk of bias tool (73.3%)	PubMed, EMBASE, Scopus, Web of Science, and Cochrane Central and Google Scholar
Ferguson et al. 2020 [116]	22 publications [27,38,43,45,46,48,50,51,52,53,57,60,63,65,66,67,70,103,106,107,117,118]	Episodic migraine, T2DM, RA, oral lichen planus, healthy, CAD, hemodialysis patients with pruritus, history of sulfur mustard intoxication, Mets, sulfur mustard-induced cutaneous complications, infertile men, NAFLD, high cholesterol or/and high CRP, overweight/obese, COPD, acute Takayasu arteritis, prediabetes	1266	30–70	P, CO	12 days–24 weeks	46–4560 mg/day of curcumin	↓ CRP (WMD = −1.55 mg/L; 95% CI −1.81 to −1.30; *p* < 0.05)	American Dietetic Association’s Quality Criteria Checklist (65.6%)	EMBASE, MEDLINE, CINAHL, Scopus, and Cochrane Library
	11 publications [24,45,46,50,63,86,103,107,114,117,118]	Episodic migraine, oral lichen planus, dyslipidemia, obese, healthy, history of sulfur mustard intoxication, Mets, sulfur mustard-induced cutaneous complications, T2DM	778	36–70	P, CO	12 days–12 weeks	80–4560 mg/day of curcumin	↓ IL-6 (WMD = −1.69 pg/mL; 95% CI −2.56 to −0.82; *p* < 0.05)		
	12 publications [24,43,46,50,63,67,86,112,114,118,119,120]	Episodic migraine, infertile men, healthy, dyslipidemia, NAFLD, overweight/obese, history of sulfur mustard intoxication, Mets, T2DM, acute Takayasu arteritis	900	30–70	P, CO	4–12	80–1140 mg/day of curcumin	↓ TNF-α (WMD = −3.13 pg/mL; 95% CI −4.62 to −1.64; *p* < 0.05)		
	4 publications [46,63,86,107]	Dyslipidemia, obese, Mets, sulfur mustard-induced cutaneous complications	539	38–56	P, CO	4	112–1140 mg/day of curcumin	↓ IL-8 (WMD = −0.54 pg/mL; 95% CI −0.82 to −0.28; *p* < 0.05)		
	3 publications [24,63,86]	Dyslipidemia, obese, Mets, history of sulfur mustard intoxication	238	38–54	P, CO	4–8	665–1140 mg/day of curcumin	↓ MCP-1 (WMD = −2.48 pg/mL; 95% CI −3.96 to −1.00; *p* < 0.05)		
	2 publications [120,121]	Healthy, episodic migraine	313	36–48	P	4–8	80–400 mg/day of curcumin	↔ ICAM-1 (WMD = −86.24 ng/mL; 95% CI −206.80 to 34.32; *p* > 0.05)		
White et al. 2019 [122]	5 publications [25,45,46,53,106]	RA, Mets, hyperlipidemia, overweight/obese with CRP ≥ 2 mg/L, ESRD	376	29–69	P, CO	4–13	20.1–500 mg/day of curcumin	↔ CRP (MD = −2.71 mg/L; 95%CI −5.73 to 0.31; *p* = 0.08)	Cochrane risk of bias tool (Supple)	PubMed, Scopus, EMBASE, Web of Science, and Cochrane Library
	6 publications [27,51,52,54,57,103]	Mets, obese hyperlipidemia, CAD, knee OA, T2DM, Hemodialysis	381	28–72	P, CO	4–12	22.1–2000 mg/day of curcumin	↔ hs-CRP (MD = −1.44 mg/L; 95%CI −2.94 to 0.06; *p* = 0.06)		
	2 publications [86,98]	Obesity, knee OA	205	42–58.6	P, CO	8–12	500–1000 mg/day of curcumin	↔ IL-1β (MD = −4.25 pg/mL; 95%CI −13.32 to 4.82; *p* = 0.36)		
	7 publications [24,45,86,93,103,114,123]	T2DM, obese, Mets, hyperlipidemia, hemodialysis, knee OA, SLE	478	20–69	P, CO	6–12	20.1–1000 mg/day of curcumin	↔ IL-6 (MD = −0.71 pg/mL; 95%CI −1.68 to 0.25; *p* = 0.15)		
	7 publications [46,86,87,103,114,123,124]	Mets, T2DM, obese, knee OA, hemodialysis, nephropathy	408	34–66	P, CO	4–12	66.3–1000 mg/day of curcumin	↔ TNF-α (MD = −1.23 pg/mL; 95%CI −3.01 to 0.55; *p* = 0.18)		
Tabrizi et al. 2019 [125]	5 publications [36,45,46,86,114]	Mets, obese, T2DM, hyperlipidemia	293	18–75	P	4–8	294 mg/day of curcumin, 300–1000 mg/day of curcuminoids, 112 mg/day of turmeric	↓ IL-6 (SMD = −2.08; 95% CI −3.90 to −0.25; *p* = 0.02)	Cochrane risk of bias tool (82.9%)	Cochrane Library, Embase, PubMed, and Web of Science
	9 publications [25,27,39,45,46,52,57,110,126]	T2DM, Mets, hyperlipidemia, overweight, obese, coronary vascular artery	832	18–75	P	3 days–36 weeks	112–2400 mg/day of turmeric, 4000 mg/day of curcuminoids, 294–1890 mg/day of curcumin	↓ hs-CRP (SMD = −0.65; 95% CI −1.20 to −0.10; *p* = 0.02)		
	5 publications [24,46,86,87,114]	Mets, overweight, obese, T2DM, diabetic nephropathy	291	18–75	P	4–8	112–1500 mg/day of turmeric, 300–1000 mg/day of curcuminoids	↔ TNF-α (SMD = −1.62; 95% CI −3.60 to 0.36; *p* = 0.10)		
Clark et al. 2019 [127]	5 publications [30,31,36,38,41]	Mets, prediabetes, obese, T2DM	652	27.1–59.37	P	6–39	200–1500 mg/day of curcumin	↑ adiponectin (WMD = 0.82 Hedges’ g; 95% CI 0.33 to 1.30; *p* ˂ 0.001) ↑ adiponectin (WMD = 1.05 Hedges’ g; 95% CI 0.64 to 1.45; *p* ˂ 0.001) for ≤ 10 weeks of follow-up period	Jadad scale (100%)	PubMed/Medline, Scopus, Web of Science, Cochrane Library, and Google scholar
Simental-Mendía et al. 2019 [128]	5 publications [31,39,41,129,130]	Mets, prediabetes, obese, T2DM	686	18–71	P	4–36	500–1500 mg/day of curcumin	↑ adiponectin (WMD = 6.47 ng/mL; 95% CI 1.85 to 11.10; *p* = 0.010)	Cochrane risk of bias tool (60.0%)	PubMed–Medline, Scopus, ISI Web of Science, and Google Scholar
Akbari et al. 2019 [131]	3 publications [24,31,32]	Mets, obese, T2DM	338	8–71	P	4–24	500–1000 mg/day of curcumin, 1500 mg/day of curcuminoids	↓ leptin (SMD = −0.97; 95% CI −1.18 to −0.75; *p* < 0.001)	Cochrane risk of bias tool (77.8%)	Cochrane Library, EMBASE, PubMed, and Web of Science
	4 publications [24,31,32,39]	Mets, obese, T2DM	572	8–71	P	4–36	500–1000 mg/day of curcumin, 1500 mg/day of curcuminoids	↑ adiponectin (SMD = 1.05; 95% CI 0.23 to 1.87; *p* = 0.01)		
Atkin et al. 2017 [132]	4 publications [31,33,36,129]	T2DM, Mets, depression	465	8–61	P	4–24	250–1000 mg/day of curcuminoids	↓ leptin (SMD = −0.69; 95% CI −1.16 to −0.23; *p* = 0.003; I^2^ = 76.53%)	Jadad scale (40%)	PubMed, Medline, Scopus, Web of Science, and Google Scholar
Derosa et al. 2016 [133]	9 publications [24,54,63,86,107,114,134,135,136]	Mets osteoarthritis, knee OA, oral lichen planus, obesity, T2DM, sulfur mustard intoxication, chronic pruritic skin lesions	609	33–69	P, CO	2–32	200–6000 mg/day of curcumin	↓ IL-6 (WMD = −0.60 pg/mL; 95% CI −1.06 to −0.14; *p* = 0.011)	Cochrane risk of bias tool (44.4%)	Medline, SCOPUS, Web of Science, and Google Scholar
Sahebkar et al. 2016 [137]	9 publications [24,63,86,87,114,123,134,138,139]	Mets, obese, T2DM, type 2 diabetic nephropathy, major depressive disorder, knee OA, sulfur mustard-exposed veterans	549	36–59	P, CO	4–12	300–1500 mg/day of curcumin	↓ TNF-α (WMD = −4.69 pg/mL; 95% CI −7.10 to −2.28; *p* < 0.001)	Cochrane risk of bias tool (42.9%)	Medline, SCOPUS, Web of Science, and Google Scholar
Sahebkar et al. 2014 [140]	7 publications [52,62,107,110,135,141,142]	Healthy, obese, dyslipidemia, chronic complications due to sulfur mustard, oral lichen planus, coronary artery bypass grafting, OA	342	25–70	P	4–12	80–6000 mg/day of curcumin	↓ CRP (MD = 6.44 mg/L; 95% CI −10.77 to −2.11; *p* = 0.004)	Jadad scale (66.7%)	Medline and SCOPUS

CAD, coronary artery disease; CHD, coronary heart disease; CI, confidence interval; CKD, chronic kidney disease; CO, crossover; COPD, chronic obstructive pulmonary disease; CRP, C-reactive protein; DFU, diabetic foot ulcer; ESRD, end-stage renal disease; HIV, human immunodeficiency virus; hs-CRP, high-sensitivity C-reactive protein; HTN, hypertension; ICAM-1, intercellular adhesion molecule-1; IGT, impaired glucose tolerance; IGTT, impaired glucose tolerance test; IL-1, interleukin 1; IL-1β, interleukin 1β; IL-6, interleukin 6; IL-8, interleukin 8; MD, mean difference; Mets, metabolic syndrome; NAFLD, non-alcoholic fatty liver disease; OA, osteoarthritis; P, parallel; PCOS, polycystic ovary syndrome; pre-HTN, pre-hypertension; PUD, peptic ulcer disease; RA, rheumatoid arthritis; RCTs, randomized controlled trials; SLE, systemic lupus erythematosus; SMD, standardized mean difference; TNF-α, tumor necrosis factor α; T2DM, type 2 diabetes mellitus; UC, ulcerative colitis; WMD, weighted mean difference. The quality of primary studies is indicated as a percentage of low risk (Cochrane risk of bias tool) or ≥3 (Jadad scale) or positive quality (American Dietetic Association’s Quality Criteria Checklist). ↑, increase; ↓, decrease; ↔ no effect.

**Table 2 nutrients-16-01728-t002:** Meta-analyses of RCTs investigating the effect of curcumin intake on antioxidant activity.

Ref.	No. of Studies Included	Health Status of Subjects	No. of Subjects	Age of Subjects (Years)	Design	Period (Weeks)	Dose	Outcomes (Effect Size)	Quality of Primary Studies	Databases
Qiu et al. 2023 [22]	2 publications [26,27]	Mets	165	34–64	P	6–12	80 mg/day of nano-curcumin, 1000 mg/day of curcuminoids	↓ MDA (MD = −2.35 µmol/L; 95% CI −4.47 to −0.24; *p* = 0.03)	Cochrane risk of bias tool (71.4%)	PubMed, SCOPUS, Cochrane Library, EMBASE, Web of Science, and China Biological Medicine
Dehzad et al. 2023 [42]	16 publications [23,30,43,49,57,68,76,77,78,79,80,82,88,143,144,145]	Mets, obese, T2DM, prostate cancer, PUD, NAFLD, hemodialysis, infertile men, β-thalassemia, hyperlipidemia, coronary elective angioplasty, DFU, healthy	857	18–70	P	4–12	80–3000 mg/day of curcumin, 2000–3000 mg/day of turmeric	↑ TAC (WMD = 0.21 mmol/L; 95% CI 0.08 to 0.33; I^2^ = 99.6%; *p* < 0.001)	Cochrane risk of bias tool (84.2%)	PubMed, Scopus, Web of Science, Cochrane Library, and Google Scholar
	18 publications [23,43,53,57,60,68,76,77,78,79,80,81,82,92,98,143,144,146]	Mets, hemodialysis, chronic pulmonary complications, T2DM, knee OA, beta-thalassemia, healthy, Takayasu arteritis, infertile men, NAFLD, HIV, hyperlipidemia, coronary elective angioplasty, DFU	1475	18–70	P	4–16	80–1500 mg/day of curcumin, 1500 mg/day of curcuminoids, 1500–3000 mg/day of turmeric	↓ MDA (WMD = −0.33 µmol/L; 95% CI −0.53 to −0.12; I^2^ = 99.6%; *p* < 0.001)		
	7 publications [60,63,64,79,95,145,147]	T2DM, solid tumor, prostate cancer, chronic pulmonary complications, PCOS, coronary elective angioplasty, chronic bronchial asthma	497	18–70	P	4–12	80–3000 mg/day of curcumin, 180–1500 mg/day of curcuminoids	↑ SOD (WMD = 20.51 u/L; 95% CI 7.35 to 33.67; I^2^ = 95.4%; *p* < 0.001)		
Jakubc-zyk et al. 2020 [148]	3 publications [43,109,143]	Overweight/obese, infertile, β-thalassemia major patients	177	13–45	P	10–12	80–1000 mg/day of curcumin	↑ TAC (SMD = 2.696; Z = 2.003; 95% CI = 0.058 to 5.335; *p* = 0.045)	Cochrane risk of bias tool (57.1%)	PubMed/MEDLINE and Embase
	3 publications [43,109,143]	Overweight/obese, infertile, β-thalassemia major patients	177	13–45	P	10–12	80–1000 mg/day of curcumin	↔ MDA (SMD = −1.579, Z = −1.714, CI = 95%, *p* = 0.086)		
Tabrizi et al. 2019 [125]	5 publications [27,57,110,114,124]	Mets, T2DM, coronary vascular artery	425	18–70	P	3 days– 8 weeks	300–4000 mg/day of curcuminoids, 2000 mg/day of turmeric	↓ MDA (SMD = −3.14; 95% CI −4.76 to −1.53; *p* < 0.001)	Cochrane risk of bias tool (82.9%)	Cochrane Library, Embase, PubMed, and Web of Science
	3 publications [27,124,149]	Mets, obese, T2DM,	260	18–65	P	4–8	1000 mg/day of curcuminoids	↔ SOD (SMD 0.34; 95% CI −1.06 to 1.74, *p* = 0.63)		
Alizadeh et al. 2019 [150]	12 publications [27,43,98,124,143,151,152,153,154,155,156,157]	Mets, knee OA, T2DM, healthy, infertile oligoasthenospermia, non-diabetic or diabetic proteinuric CKD, chronic gastritis, β-thalassemia, hemodialysis, sulfur mustard Iraq–Iran war chronic pulmonary complications, solid cancer	1005	20–75	P	1–16	80–1000 mg/day of curcumin, 90–1500 mg/day of curcuminoids, 1500–2100 mg/day of turmeric	↓ MDA (SMD = −0.46; 95% CI −0.68 to −0.25)	Cochrane risk of bias tool (59.0%)	PubMed, Embase, Cochrane Central, Scopus, and Google Scholar
	7 publications [62,124,141,145,155,156,158]	Healthy, prostate cancer treated with radiotherapy, veterans of the Iraq–Iran war with chronic pruritus, sulfur mustard Iraq–Iran war chronic pulmonary complications, solid cancer, Mets, knee OA, T2DM	618	35–79	P	4–12	900–3000 mg/day of curcuminoids	↑ SOD (SMD = 0.82; 95% CI 0.27 to 1.38)		
	5 publications [62,141,143,145,155]	Healthy, prostate cancer treated with radiotherapy, β-thalassemia, veterans of the Iraq–Iran war with chronic pruritus, solid cancer	322	20–79	P	4–12	900–3000 mg/day of curcuminoids	↑ CAT (SMD = 10.26; 95% CI 0.92 to 19.61)		
	5 publications [62,141,145,151,158]	Non-diabetic or diabetic proteinuric CKD, prostate cancer treated with radiotherapy, veterans of the Iraq-Iran war with chronic pruritus, occupational stress-related anxiety and fatigue	357	26–79	P	4–12	320–782 mg/day of curcumin, 1000–3000 mg/day of curcuminoids	↑ GPx (SMD = 8.90; 95% CI 6.62 to 11.19)		
	6 publications [43,88,143,145,152,157]	PUD, infertile oligoasthenospermia, prostate cancer treated with radiotherapy, β-thalassemia, chronic gastritis, healthy	330	22–79	P	1–12	80–1500 mg/day of curcumin, 90–3000 mg/day of curcuminoids	↔ TAC (SMD = 0.30; 95% CI −0.20 to 0.81)		
	1 publication [151]	Non-diabetic or diabetic proteinuric CKD	101	37–57	P	8	320 mg/day of curcumin	↔ GR (SMD = −0.01; 95% CI −0.03 to 0.02)		
Qin et al. 2018 [159]	7 publications [27,57,98,114,124,153,156]	T2DM, OA, Mets, end-stage renal disease	525	34–67	P	4–8	46–1000 mg/day of curcumin	↓ MDA (SMD = −0.769; 95% CI −1.059 to −0.478)	Cochrane risk of bias tool (54.2%)	PubMed, EMBASE, Web of Science, Medline Ovid, Books^@^Ovid, Journals^@^Ovid, and Cochrane Library
	4 publications [27,124,141,156]	T2DM, OA, Mets, chronic pruritic skin lesions	341	34–67	P	4–8	1000–1500 mg/day of curcumin	↑ SOD (SMD = 1.084; 95% CI 0.487 to 1.680)		
	2 publications [57,153]	T2DM, end-stage renal disease	108	32–67	P	4–8	46–66.3 mg/day of curcumin	↔ GPx (SMD = 0.13; 95% CI −0.25 to 0.51)		

CAT, chloramphenicol acetyltransferase; CKD, chronic kidney disease; DFU, diabetic foot ulcer; GPx, glutathione peroxidase; GR, glutathione reductase; HIV, human immunodeficiency virus; MD, mean difference; MDA, malondialdehyde; Mets, metabolic syndrome; NAFLD, non-alcoholic fatty liver disease; OA, osteoarthritis; P, parallel; PCOS, polycystic ovary syndrome; PUD, peptic ulcer disease; RCTs, randomized controlled trials; SMD, standard difference in means; SOD, superoxide dismutase; TAC, total antioxidant capacity; T2DM, type 2 diabetes mellitus; WMD, weighted mean differences. The quality of primary studies is indicated as a percentage of low risk (Cochrane risk of bias tool). ↑, increase; ↓, decrease; ↔ no effect.

**Table 3 nutrients-16-01728-t003:** Meta-analyses of RCTs investigating the effect of curcumin intake on glucose control.

Ref.	No. of Studies Included	Health Status of Subjects	No. of Subjects	Age of Subjects (Years)	Design	Period (Weeks)	Dose	Outcomes (Effect Size)	Quality of Primary Studies	Databases
Lukkunaprasit et al. 2023 [160]	3 publications [50,161,162]	MAFLD	231	26–38	P	8–12	80–500 mg/day of curcumin	↔ HbA1c (MD = −0.24; 95% CI −0.66 to 0.18; *p* > 0.05)	Cochrane risk of bias tool (68.8%)	Medline and SCOPUS
	12 publications [34,35,50,146,161,162,163,164,165,166,167,168]	MAFLD	710	18–72	P	8–24	80–1500 mg/day of curcumin, 2000–3000 mg/day of turmeric	↓ FBG (MD = −2.05 mg/dL; 95% CI −3.08 to −1.01; *p* < 0.05)		
Qiu et al. 2023 [22]	9 publications [23,25,27,61,126,169,170,171,172]	Mets	576	28–81	P	4–12	80–1500 mg/day of curcumin, 1000 mg/day of curcuminoids, 2400 mg/day of turmeric	↓ FBG (MD = −8.6 mg/dL; 95% CI −15.45 to −1.75; *p* = 0.01)	Cochrane risk of bias tool (71.4%)	PubMed, SCOPUS, Cochrane Library, EMBASE, Web of Science, and China Biological Medicine
Różański et al. 2023 [173]	14 publications [34,35,50,55,161,164,165,166,174,175,176,177,178,179]	MAFLD	847	18–70	P	8–12	70–1500 mg/day of curcuminoids, 80–1000 mg/day of curcumin, 3000 mg/day of turmeric	↓ Insulin (MD = −1.1430 μIU/mL; 95% CI −1.5439 to −0.7421; *p* < 0.0001) ↓ HOMA-IR (MD = −0.2884; 95%CI −0.3950 to −0.1817; *p* < 0.0001)	Cochrane risk of bias tool (90.8%)	PubMed, Web of Science, and Scopus
Ngu et al. 2022 [180]	3 publications [50,177,181]	NAFLD	238	18–70	P	8–12	80–1500 mg/day of curcumin	↔ QUICKI (MD = 0.01; 95% CI −0.00 to 0.02; *p* = 0.30)	Cochrane risk of bias tool (79.5%)	Cochrane Central Register of Controlled Trials, and PubMed
Sun et al. 2022 [100]	23 publications [23,25,27,39,45,49,60,61,70,76,80,87,101,126,169,170,182,183,184,185,186,187,188]	T2DM, obese, nephropathy, proteinuria, hemodialysis, DFU, IFG/IGT, Mets, hyperlipidemia	1847	35–68	P, CO	4–36	80–2400 mg/day of curcumin	↓ FBG (WMD = −0.50 mg/dL; 95% CI −0.72 to −0.28)	Cochrane risk of bias tool (69.3%) Jadad scale (77.0%)	PubMed, Embase, Web of Science, CNKI, Wanfang and CBM
	16 publications [27,29,39,49,60,70,76,80,101,126,182,183,185,186,187,188]	T2DM, obese, proteinuria, hemodialysis, DFU, IFG/IGT, Mets, hyperlipidemia	1224	35–70	P, CO	8–36	80–2100 mg/day of curcumin	↓ Hb1Ac (WMD = −0.42%; 95% CI −0.57 to −0.26)		
	5 publications [45,76,80,187,188]	T2DM, IGT, hyperlipidemia, hemodialysis, DFU	247	35–63	P, CO	6–17	80–500 mg/day of curcumin	↓ Insulin (WMD = −1.70 μIU/mL; 95% CI −2.03 to −1.38)		
	11 publications [39,45,49,60,70,76,80,101,183,188,189]	T2DM, obese, hyperlipidemia, hemodialysis, IFG, IGT, DFU	874	42–63	P, CO	6–36	80–2100 mg/day of curcumin	↓ HOMA-IR (WMD = −0.71; 95% CI −1.11 to −0.31)		
Tian et al. 2022 [190]	9 publications [38,49,57,87,101,114,182,183,191]	T2DM	604	41–61	P	4–12	80–1500 mg/day of curcumin, 300–1000 mg/day of curcuminoids, 1500–2100 mg/day of turmeric	↓ FBG (WMD = −8.85 mg/dL; 95% CI −14.4 to −3.29; *p* = 0.002)	Cochrane risk of bias tool (57.1%)	PubMed, EMBASE, Web of Science, and Cochrane Library
	8 publications [38,49,57,101,114,182,183,191]	T2DM	564	41–61	P	4–12	80–1500 mg/day of curcumin, 300–1000 mg/day of curcuminoids, 2100 mg/day of turmeric	↓ HbA1c (WMD = −0.54; 95% CI −0.81 to −0.27; *p* ≤ 0.001)		
Nouri et al. 2022 [192]	4 publications [58,73,193,194]	PCOS	198	28–31	P	6–12	93.34–1500 mg/day of curcumin	↓ FBG (ES = −3.62 mg/dL; 95% CI −5.65 to −1.58; *p* < 0.001) ↓ Insulin (ES = −1.67 mU/mL; 95% CI −3.06 to −0.28; *p* = 0.018) ↓ HOMA-IR (ES = −0.42; 95% CI −0.76 to −0.09; *p* < 0.01)	Cochrane risk of bias tool (75.0%)	PubMed, Scopus, and ISI Web of Science
Khalili et al. 2022 [195]	9 publications [34,50,161,164,165,166,177,179,196]	NAFLD	683	42–67	P	4–12	50–2000 mg/day of curcumin	↓ FBG (SMD = −0.28; 95% CI −0.46 to −0.09; *p* = 0.003)	Cochrane risk of bias tool (65.5%)	Web of Science, Science Direct, PubMed, and Embase
Shen et al. 2022 [99]	7 publications [58,73,193,194,197,198,199]	PCOS	447	20–38	P	6–24	80–1500 mg/day of curcumin	↓ FBG (WMD = −3.618; 95% CI −5.165 to −2.071; *p* < 0.001) ↓ Insulin (WMD = −1.834; 95% CI −2.701 to −0.968; *p* < 0.001) ↓ HOMA-IR (WMD = −0.565; 95% CI −0.779 to −0.351; *p* < 0.001)	Cochrane risk of bias tool (73.5%)	PubMed, Embase, Cochrane Library, Web of Science, Scopus, Clinical Trials, Chinese Clinical Trial Registry, Chinese Biomedical Literature Database, Chinese National Knowledge Infrastructure, VIP database, and Wanfang Database
	4 publications [58,193,194,197]	PCOS	266	23–38	P	6–12	80–1500 mg/day of curcumin	↑ QUICKI (WMD = 0.011; 95% CI 0.005 to 0.017; *p* = 0.001)		
Ashtary-Larky et al. 2021 [111]	7 publications [23,50,71,76,171,182,185]	Mets, NAFLD, T2DM, hemodialysis	417	36–68	P	6–12	40–120 mg/day of nano-curcumin	↓ FBG (WMD = −18.14 mg/dL; 95% CI −29.31 to −6.97; *p* = 0.001)	Cochrane risk of bias tool (55.6%)	PubMed, Scopus, Embase, and Web of Science
	3 publications [50,76,171]	Mets, NAFLD, T2DM, hemodialysis	180	36–68	P	12	40–80 mg/day of nano-curcumin	↓ Insulin (WMD = −1.21 mg/dL; 95% CI −1.43 to −1.00; *p* < 0.001)		
	4 publications [50,171,182,185]	Mets, NAFLD, T2DM	277	36–68	P	8–12	40–80 mg/day of nano-curcumin	↓ HbA1c (WMD = −0.66 mg/dL; 95% CI −1.41 to −0.08; *p* < 0.001)		
	3 publications [50,76,171]	Mets, NAFLD, T2DM, hemodialysis	180	36–68	P	12	40–80 mg/day of nano-curcumin	↓ HOMA-IR (WMD = −0.28 mg/dL; 95% CI −0.33 to −0.23; *p* < 0.001)		
Abdelazeem et al. 2021 [115]	5 publications [58,73,193,194,197]	PCOS	296	24–39	P	6–12	80–1500 mg/day of curcumin	↓ FBG (MD = −3.67; 95% CI −5.25 to −2.08; *p* < 0.00001) ↓ Insulin (MD = −1.91; 95% CI −2.97 to −0.84; *p* = 0.0005) ↓ HOMA-IR (MD = −0.55; 95% CI −0.83 to −0.27; *p* = 0.0001)	Cochrane risk of bias tool (73.3%)	PubMed, EMBASE, Scopus, Web of Science, Cochrane Central, and Google Scholar
	4 publications [58,193,194,197]	PCOS	266	24–36	P	6–12	80–1500 mg/day of curcumin	↓ QUICKI (MD = 0.01; 95% CI 0.00 to 0.02; *p* = 0.0005)		
Zhang et al. 2021 [200]	5 publications [31,60,70,183,185]	T2DM	524	34–70	P	8–24	80–1500 mg/day of curcumin, 300–1500 mg/day of curcuminoids	↓ HbA1c (WMD = −0.70; 95% CI −0.87 to −0.54; *p* < 0.0001)	Cochrane risk of bias tool (69.4%)	Chinese database (CNKI, Wan Fang, VIP, CBM), PubMed, EMBASE, Cochrane Library, Web of Science, Medline Complete and, ClinicalTrials.gov
	1 publication [60]	T2DM	100	34–51	P	12	1000 mg/day of curcuminoids	↓ HOMA-IR in the Middle East subgroup (WMD = −0.60; 95% CI −0.74 to −0.46; *p* < 0.00001)		
	2 publications [31,183]	T2DM	313	34–65	P	12–24	1500 mg/day of curcumin, 300 mg/day of curcuminoids	↓ HOMA-IR in the Asia subgroup (WMD = −2.41; 95% CI −4.44 to −0.39; *p* = 0.02)		
	6 publications [31,60,70,87,183,185]	T2DM	564	34–70	P	8–24	80–1500 mg/day of curcumin, 300–1500 mg/day of curcuminoids, 1500 mg/day of turmeric	↓ FBG in the Asia subgroup (SMD = −0.57; 95% CI −0.79 to −0.36; *p* < 0.00001) ↔ FBG in the Middle East subgroup		
Altobelli et al. 2021 [201]	5 publications [49,114,144,182,183]	T2DM	333	18–80	P	8–12	80–1500 mg/day of curcumin, 300 mg/day of curcuminoids, 2100 mg/day of turmeric	↓ HbA1c (ES = −0.42; 95% CI −0.77 to −0.11; *p* = 0.008)	Cochrane risk of bias tool (93.8%)	Medline, EMBASE, Scopus, ClinicalTrials.gov, Web of Science, and Cochrane Library
	4 publications [31,49,144,183]	T2DM	432	18–70	P	8–24	1500 mg/day of curcumin, 250–300 mg/day of curcuminoids, 2100 mg/day of turmeric	↓ HOMA-IR (ES = −0.41; 95% CI −0.60 to −0.22; *p* < 0.001)		
Chien et al. 2021 [202]	3 publications [58,193,194]	PCOS	168	24–39	P	6–12	500–1500 mg/day of curcumin	↓ FBG (MD = −2.77; 95% CI −4.16 to −1.38; *p* < 0.001) ↓ Insulin (MD = −1.33; 95% CI −2.18 to −0.49; *p* = 0.002) ↓ HOMA-IR (MD = −0.32; 95% CI −0.52 to −0.12; *p* = 0.002) ↓ QUICKI (MD = 0.010; 95% CI 0.003 to 0.018; *p* = 0.005)	Cochrane risk of bias tool (72.2%)	PubMed, Embase, Scopus, Web of Science, and Cochrane Library
Jalali et al. 2020 [203]	6 publications [34,50,161,164,165,177]	NAFLD	407	33–58	P	8–12	50–1425 mg/day of curcumin	↔ FBG (MD = −0.22; 95% CI −0.42 to −0.02; *p* = 0.313)	Jadad scale (88.9%)	PubMed, Embase, Scopus, Web of Science, and Cochrane Library
	3 publications [50,165,177]	NAFLD	211	36–58	P	8–12	80–1425 mg/day of curcumin	↓HOMA-IR (MD = −0.37; 95% CI −0.70 to −0.03; *p* = 0.000) ↓ Insulin (MD = −0.49; 95% CI −0.81 to −0.16; *p* = 0.000)		
Azhda-ri et al. 2019 [204]	5 publications [25,27,61,126,172]	Mets	359	38–60	P	6–12	1000–1890 mg/day of curcumin, 1000 mg/day of curcuminoids, 2400 mg/day of turmeric	↓ FBG (WMD = −9.18 mg/dL; 95% CI −16.70 to −1.66; *p* = 0.01)	Cochrane risk of bias tool (57.1%)	Web of Science, Medline, Google Scholar, Scopus, Cochrane, and CINAHL
Huang et al. 2019 [205]	13 publications [25,35,39,45,51,57,114,126,161,177,182,183,206]	T2DM, NAFLD, Mets, hyperlipidemia, CAD, high ALT	1064	30–72	P, CO	4–12	70–1890 mg/day of curcumin, 294–2000 mg/day of curcuminoids, 2400–3000 mg/day of turmeric	↓ FBG (SMD = −0.382; 95% CI −0.654 to −0.111 mg/dL; *p* = 0.006)	Cochrane risk of bias tool (83.7%)	PubMed, Cochrane Library, Web of Science, and Embase
	7 publications [31,35,39,45,57,177,183]	T2DM, NAFLD, Mets, hyperlipidemia	821	30–72	P, CO	4–12	1000 mg/day of curcumin, 294–1500 mg/day of curcuminoids, 2000–3000 mg/day of turmeric	↓ HOMA-IR (SMD = −0.351; 95% CI −0.615 to −0.087; *p* = 0.009)		
	8 publications [39,57,114,126,161,177,182,183]	T2DM, NAFLD, Mets	731	40–74	P	4–12	80–1890 mg/day of curcumin, 300–1500 mg/day of curcuminoids, 2000 mg/day of turmeric	↓ HbA1c (SMD = −0.370; 95% CI −0.631% to −0.110%; *p* = 0.005)		
	5 publications [35,39,45,57,177]	T2DM, NAFLD, Mets, hyperlipidemia	507	31–71	P, CO	4–12	1000 mg/day of curcumin, 294–1500 mg/day of curcuminoids, 2000–3000 mg/day of turmeric	↔ Insulin (SMD = −0.058; 95% CI −0.352 to 0.235; *p* = 0.697)		
Tabrizi et al. 2018 [207]	19 publications [25,32,39,45,46,51,57,87,101,114,126,129,161,178,182,183,206,208,209]	T2DM, hyperlipidemia, Mets, obese, diabetic nephropathy, NAFLD, CAD	1299	10–73	P	4–24	70–2000 mg/day of curcumin, 300–1000 mg/day of curcuminoids, 112–2400 mg/day of turmeric	↓ FBG (SMD = −0.78; 95% CI −1.20 to −0.37; *p* < 0.001)	Cochrane risk of bias tool (76.9%)	Cochrane Library, EMBASE, MEDLINE, and Web of Science
	7 publications [32,39,45,57,129,178,208]	T2DM, obese, hyperlipidemia, NAFLD	552	18–68	P	4–36	294 mg/day of curcumin, 500–1000 mg/day of curcuminoids, 2000–2100 mg/day of turmeric	↑ Insulin (SMD = 0.92; 95% CI 0.06 to 1.78; *p* = 0.036)		
	8 publications [31,32,39,45,57,178,183,208]	T2DM, obese, hyperlipidemia, NAFLD	836	10–65	P	4–36	294 mg/day of curcumin, 500–1000 mg/day of curcuminoids, 300–2100 mg/day of turmeric	↓ HOMA-IR (SMD = −0.91; 95% CI −1.52 to −0.31; *p* = 0.003)		
	10 publications [39,57,101,114,126,161,178,182,183,208]	T2DM, Mets, obese, hyperlipidemia, NAFLD	906	18–73	P	4–36	70–1890 mg/day of curcumin, 300–1000 mg/day of curcuminoids, 300–2100 mg/day of turmeric	↓ HbA1c (SMD = −0.92; 95% CI −1.37 to −0.47; *p* < 0.001)		

ALT, alanine aminotransferase; CAD, coronary artery disease; CI, confidence interval; CO, crossover; DFU, diabetic foot ulcer; ES, effect size; FBG, fasting blood glucose; HbA1c, glycated hemoglobin; HOMA-IR, homeostatic model assessment for insulin resistance; IFG, impaired fasting glucose; IGT, impaired glucose tolerance; MD, mean difference; Mets, metabolic syndrome; MAFLD, metabolic (dysfunction)-associated fatty liver disease; NAFLD, non-alcoholic fatty liver disease; P, parallel, PCOS, polycystic ovary syndrome; QUICKI, quantitative insulin sensitivity check index; RCTs, randomized controlled trials; SMD, standardized mean difference; T2DM, type 2 diabetes mellitus; WMD, weighted mean difference. The quality of primary studies is indicated as a percentage of low risk (Cochrane risk of bias tool) or ≥3 (Jadad scale). ↑, increase; ↓, decrease; ↔ no effect.

**Table 4 nutrients-16-01728-t004:** Meta-analyses of RCTs investigating the effect of curcumin intake on lipid profile changes.

Ref.	No. of Studies Included	Health Status of Subjects	No. of Subjects	Age of Subjects (Years)	Design	Period (Weeks)	Dose	Outcomes (Effect Size)	Quality of Primary Studies	Databases
Lukkunaprasit et al. 2023 [160]	11 publications [34,50,146,161,162,163,164,165,166,167,168]	MAFLD	335	21–72	P	8–24	80–1500 mg/day of curcumin, 2000 mg/day of turmeric	↓ TC (MD = −8.18 mg/dL; 95% CI −14.07 to −2.29; *p* < 0.05) ↔ TG (MD = −12.03 mg/dL; 95% CI −24.36 to 0.30; *p* = 0.27) ↔ LDL-C (MD = −4.35 mg/dL; 95% CI −10.93 to 2.23; *p* = 0.73)	Cochrane risk of bias tool (68.8%)	Medline and SCOPUS
	10 publications [34,50,146,161,162,164,165,166,167,168]	MAFLD	325	28–59	P	8–12	80–1500 mg/day of curcumin, 2000 mg/day of turmeric	↔ HDL-C (MD = 0.80 mg/dL; 95% CI −2.92 to 4.52; *p* = 0.41)		
Qiu et al. 2023 [22]	8 publications [23,25,61,126,169,170,171,172]	Mets	476	28–81	P	4–12	80–1890 mg/day of curcumin, 2400 mg/day of turmeric	↔ TG (MD = 1.28 mg/dL; 95% CI −3.75 to 6.30; *p* = 0.62) ↓ HDL-C (MD = 4.98 mg/dL; 95% CI 2.58 to 7.38; *p* < 0.0001)	Cochrane risk of bias tool (71.4%)	PubMed, SCOPUS, Cochrane Library, EMBASE, Web of Science, and China Biological Medicine
Rozanski et al. 2023 [173]	14 publications [34,35,50,55,161,164,165,166,174,175,176,177,178,179]	MAFLD	847	18–70	P	8–12	70–1500 mg/day of curcuminoids, 80–1000 mg/day of curcumin, 3000 mg/day of turmeric	↓ TG (MD = −12.6001 mg/dL; 95% CI −22.9311 to −2.2692; *p* = 0.0168) ↓ TC (MD = −15.7896 mg/dL; 95% CI −28.2686 to −3.3106; *p* = 0.0131) ↓ LDL-C (MD = −14.1699 mg/dL; 95% CI −25.9117 to −2.4281; *p* = 0.018)	Cochrane risk of bias tool (90.8%)	PubMed, Web of Science, and Scopus
Ngu et al. 2022 [180]	10 publications [34,50,161,162,163,164,165,166,177,196]	NAFLD	771	18–70	P	8–24	50–1500 mg/day of curcumin, 2000 mg/day of turmeric	↓ TC (MD = −11.86; 95% CI −19.25 to −4.46; *p* = 0.002) ↔ LDL-C (MD = −8.78; 95% CI −18.74 to 1.18; *p* = 0.08) ↓ TG (MD = −13.00; 95% CI −27.94 to 1.94; *p* = 0.09)	Cochrane risk of bias tool (79.5%)	Cochrane Central Register of Controlled Trials and PubMed
	9 publications [34,50,161,162,164,165,166,177,196]	NAFLD	638	18–70	P	8–12	50–1500 mg/day of curcumin, 2000 mg/day of turmeric	↔ HDL-C (MD = 0.43; 95% CI −3.88 to 4.74; *p* = 0.85)		
Shen et al. 2022 [99]	5 publications [58,73,193,197,199]	PCOS	380	20–38	P	6–12	80–1500 mg/day of curcumin	↓ TC (WMD = −15.591; 95% CI −27.908 to −3.273; *p* = 0.013) ↔ TG (WMD = −8.889; 95% CI −27.246 to 9.468; *p* = 0.343) ↔ LDL-C (WMD = −6.427; 95% CI −17.343 to 4.489; *p* = 0.249) ↔ HDL-C (WMD = 3.713; 95% CI −0.786 to 8.211; *p* = 0.106)	Cochrane risk of bias tool (73.5%)	PubMed, Embase, Cochrane Library, Web of Science, Scopus, Clinical Trials, Chinese Clinical Trial Registry, Chinese Biomedical Literature Database, Chinese National Knowledge Infrastructure, VIP database, and Wanfang Database
Tian et al. 2022 [190]	9 publications [38,49,57,87,101,114,182,183,191]	T2DM	604	41–61	P	4–12	80–1500 mg/day of curcumin, 300–1000 mg/day of curcuminoids, 1500–2100 mg/day of turmeric	↓ TG (WMD = −18.97 mg/dL; 95% CI −36.47 to −1.47; *p* = 0.03)	Cochrane risk of bias tool (57.1%)	PubMed, EMBASE, Web of Science, and Cochrane Library
	9 publications [38,49,57,87,101,114,182,183,191]	T2DM	604	41–61	P	4–12	80–1500 mg/day of curcumin, 300–1000 mg/day of curcuminoids, 1500–2100 mg/day of turmeric	↓ TC (WMD = −8.91 mg/dL; 95% CI −14.18 to −3.63; *p* = 0.001)		
	9 publications [38,49,57,87,101,114,182,183,191]	T2DM	604	41–61	P	4–12	80–1500 mg/day of curcumin, 300–1000 mg/day of curcuminoids, 1500–2100 mg/day of turmeric	↔ LDL-C (WMD = −4.01 mg/dL; 95% CI −10.96 to 2.95; *p* = 0.259)		
	9 publications [38,49,57,87,101,114,182,183,191]	T2DM	604	41–61	P	4–12	80–1500 mg/day of curcumin, 300–1000 mg/day of curcuminoids, 1500–2100 mg/day of turmeric	↔ HDL-C (WMD = 0.32 mg/dL; 95% CI −0.74 to 1.37; *p* = 0.557)		
Sun et al. 2022 [100]	16 publications [25,38,45,49,61,70,76,80,126,169,170,184,186,188,209,210]	Mets, hyperlipidemia, hypercholesterolemia, T2DM, DFU, IGT, IFG, proteinuria, hemodialysis	1274	38–68	P, CO	4–16	80–2400 mg/day of curcumin	↔ TC (WMD = −2.45 mg/dL; 95% CI −9.00 to 4.09; *p* = 0.063)	Cochrane risk of bias tool (69.3%) Jadad scale (77.0%)	PubMed, Embase, Web of Science, CNKI, Wanfang, and CBM
	19 publications [23,25,29,38,45,49,61,70,76,126,169,170,182,183,184,186,188,209,210]	Mets, obese, hyperlipidemia, hypercholesterolemia, T2DM, DFU, IGT, IFG, proteinuria, hemodialysis	1561	38–70	P, CO	4–24	80–2100 mg/day of curcumin	↔ TG (WMD = −10.50 mg/dL; 95% CI −23.18 to 2.18; *p* = 0.105)		
	19 publications [23,25,29,38,45,49,61,70,76,126,169,170,182,183,184,186,188,209,210]	Mets, obese, hyperlipidemia, hypercholesterolemia, T2DM, DFU, IGT, IFG, proteinuria, hemodialysis	1561	38–70	P, CO	4–24	80–2100 mg/day of curcumin	↑ HDL-C (WMD = 1.73 mg/dL; 95% CI 0.78 to 2.68; *p* < 0.001)		
	17 publications [25,29,38,45,49,61,70,76,80,169,170,183,184,186,188,209,210]	Mets, obese hyperlipidemia, hypercholesterolemia, T2DM, DFU, IGT, IFG, proteinuria, hemodialysis	1256	39–70	P, CO	4–24	80–2100 mg/day of curcumin	↔ LDL-C (WMD = −3.99 mg/dL; 95% CI −8.68 to 0.68; *p* = 0.094)		
Saeedi et al. 2022 [211]	10 publications [25,51,101,126,164,165,177,184,191,209]	T2DM, CVD, Mets, dyslipidemia, NAFLD	726	18–65	P	4–12	50–2000 mg/day of curcumin, 1000 mg/day of curcuminoids, 2100 mg/day of turmeric	↔ TG (SMD = −0.05; 95% CI −0.20 to 0.11; *p* = 0.56) ↔ TC (SMD = −0.21; 95% CI −0.55 to 0.13; *p* = 0.22) ↔ LDL-C (SMD = −0.17; 95% CI −0.43 to 0.09; *p* = 0.82) ↔ HDL-C (SMD = −0.04; 95% CI −0.36 to 0.29; *p* = 0.82)	Cochrane risk of bias tool (not indicate)	Ovid-Medline, Web of Science, Scopus, Cochrane, Embase, and ProQuest
Futuhi et al. 2022 [102]	5 publications [71,72,76,151,212]	ESRD, diabetic proteinuric non-diabetic proteinuric CKD, hemodialysis	276	23–69	P	8–12	80–500 mg/day of curcumin, 2500 mg/day of turmeric	↓ TC (WMD = −13.77 mg/dL; 95% CI −26.77 to −0.77; *p* = 0.04)	Cochrane risk of bias tool (73.0%)	PubMed, Scopus, Cochrane Library, and Web of Science
	5 publications [71,72,76,151,212]	ESRD, diabetic proteinuric non-diabetic proteinuric CKD, hemodialysis	276	23–69	P	8–12	80–500 mg/day of curcumin, 2500 mg/day of turmeric	↔ TG (WMD = −6.37 mg/dL; 95% CI −26.59 to 13.85; *p* = 0.54)		
	4 publications [71,72,76,212]	ESRD, hemodialysis	175	23–69	P	12	80–500 mg/day of curcumin, 2500 mg/day of turmeric	↔ LDL-C (WMD = −5.65 mg/dL; 95% CI −20.81 to 9.50; *p* = 0.46)		
	4 publications [71,72,76,212]	ESRD, hemodialysis	175	23–69	P	12	80–500 mg/day of curcumin, 2500 mg/day of turmeric	↔ HDL-C (WMD = 0.16 mg/dL; 95% CI −2.55 to 2.88; *p* = 0.91)		
Khalili et al. 2022 [195]	8 publications [34,50,161,165,166,177,179,196]	NAFLD	638	42–67	P	4–12	50–2000 mg/day of curcumin	↓ TG (SMD = −0.49; 95% CI −0.71 to −0.27; *p* = 0.000)	Cochrane risk of bias tool (65.5%)	Web of Science, Science Direct, PubMed, and Embase
	9 publications [34,50,161,164,165,166,177,179,196]	NAFLD	683	42–67	P	4–12	50–2000 mg/day of curcumin	↔ LDL-C (SMD = 0.48; 95% CI −0.97 to 0.01; *p* = 0.053)		
	9 publications [34,50,161,164,165,166,177,179,196]	NAFLD	683	42–67	P	4–12	50–2000 mg/day of curcumin	↓ TC (SMD = 0.81; 95% CI −1.34 to −0.27; *p* = 0.003)		
	8 publications [34,161,164,165,166,177,179,196]	NAFLD	599	42–67	P	4–12	50–2000 mg/day of curcumin	↔ HDL-C (SMD = 0.03; 95% CI −0.38 to 0.44; *p* = 0.886)		
Zhang et al. 2021 [200]	2 publications [31,183]	T2DM	313	18–65	P	12–24	1500 mg/day of curcumin, 300 mg/day of curcuminoids	In the Asia subgroup, ↓ TC (WMD = −23.45 mg/dL; 95% CI −40.04 to −6.84; *p* = 0.006) ↓ TG (WMD = −54.14 mg/dL; 95% CI −95.71 to −12.57; *p* = 0.01)	Cochrane risk of bias tool (69.4%)	Chinese database (CNKI, Wan Fang, VIP, CBM), PubMed, EMBASE, Cochrane Library, Web of Science, Medline Complete, and ClinicalTrials.gov
	2 publications [87,101]	T2DM	140	34–62	P	8–12	1000 mg/day of curcuminoids, 1500 mg/day of turmeric	In the Middle East subgroup, ↔ TC (WMD = 22.91; 95% CI −16.94 to 62.75; *p* = 0.26) ↔ TG (WMD = −4.56; 95% CI −19.28 to 10.16; *p* = 0.54)		
	2 publications [31,183]	T2DM	313	18–65	P	12–24	1500 mg/day of curcumin, 300 mg/day of curcuminoids	↓ LDL-C in patients with T2DM in the Asia subgroup only (WMD = −20.85 mg/dL; 95% CI −28.78 to −12.92; *p* < 0.00001)		
	5 publications [31,38,87,119,183]	T2DM	497	34–70	P	8–24	1500 mg/day of curcumin, 300–1000 mg/day of curcuminoids, 1500 mg/day of turmeric	↔ HDL-C (WMD = 2.26 mg/dL; 95% CI −2.03 to 6.55; *p* = 0.30)		
Altobelli et al. 2021 [201]	5 publications [38,49,114,182,183]	T2DM	476	18–80	P	8–12	80–1500 mg/day of curcumin, 300 mg/day of curcuminoids, 2100 mg/day of turmeric	↓ TG (ES = −0.57; 95% CI −0.83 to −0.31; *p* < 0.001)	Cochrane risk of bias tool (93.8%)	Medline, EMBASE, Scopus, ClinicalTrials.gov, Web of Science, and Cochrane Library
	5 publications [38,49,114,182,183]	T2DM	312	18–80	P	8–12	80–1500 mg/day of curcumin, 300 mg/day of curcuminoids, 2100 mg/day of turmeric	↓ TC (ES = −0.30; 95% CI −0.53 to −0.07; *p* < 0.001)		
	5 publications [38,49,114,182,183]	T2DM	333	18–80	P	8–12	80–1500 mg/day of curcumin, 300 mg/day of curcuminoids, 2100 mg/day of turmeric	↔ HDL-C (ES = 0.22; 95% CI −0.08 to 0.52; *p* = 0.143)		
	5 publications [31,38,49,114,183]	T2DM	300	18–80	P	8–12	80–1500 mg/day of curcumin, 300 mg/day of curcuminoids, 2100 mg/day of turmeric	↓ LDL-C (ES = −0.28; 95% CI −0.52 to −0.04; *p* = 0.021)		
Abdelazeem et al. 2021 [115]	4 publications [58,73,193,197]	PCOS	229	24–35	P	6–12	80–1000 mg/day of curcumin	↔ TG (MD = −10.18; 95% CI −31.20 to 10.83; *p* = 0.34) ↔ LDL-C (MD = −8.53; 95% CI −22.97 to 5.91; *p* = 0.25) ↔ HDL-C (MD = 4; 95% CI −1.74 to 9.75; *p* = 0.17) ↓ TC (MD = −15.55; 95% CI −30.33 to −0.76; *p* < 0.04)	Cochrane risk of bias tool (73.3%)	PubMed, EMBASE, Scopus, Web of Science, Cochrane Central, and Google Scholar
Chien et al. 2021 [202]	2 publications [58,193]	PCOS	101	24–35	P	6–12	1000–1500 mg/day of curcumin	↓ HDL (MD = 1.92; 95% CI 0.33 to 3.51; *p* = 0.018) ↓ TC (MD = −12.45; 95% CI −22.05 to −2.85; *p* = 0.011) ↔ LDL (MD = −6.02; 95% CI −26.66 to 14.62; *p* = 0.567) ↔ TG (MD = 8.22; 95% CI −26.10 to 42.53; *p* = 0.639)	Cochrane risk of bias tool (72.2%)	PubMed, Embase, Scopus, Web of Science, and Cochrane Library
Ashtary-Larky et al. 2021 [111]	6 publications [23,50,71,76,171,182]	NAFLD, T2DM, Mets, hemodialysis	337	36–68	P	6–12	40–120 mg/day of nano-curcumin	↔ TG (WMD = −9.76 mg/dL; 95% CI −32.71 to 13.17; *p* = 0.404)	Cochrane risk of bias tool (55.6%)	PubMed, Scopus, Embase, and Web of Science
	5 publications [50,71,76,171,182]	NAFLD, T2DM, Mets, hemodialysis	304	36–68	P	12	40–120 mg/day of nano-curcumin	↔ TC (WMD = −3.34 mg/dL; 95% CI −14.43 to 7.73; *p* = 0.554)		
	5 publications [50,71,76,171,182]	NAFLD, T2DM, Mets, hemodialysis	304	36–68	P	12	40–120 mg/day of nano-curcumin	↔ LDL-C (WMD = −3.59 mg/dL; 95% CI −15.74 to 8.56; *p* = 0.562)		
	6 publications [23,50,71,76,171,182]	NAFLD, T2DM, Mets, hemodialysis	337	36–68	P	6–12	40–120 mg/day of nano-curcumin	↑ HDL-C (WMD = 5.77 mg/dL; 95% CI 2.90 to 8.64; *p* < 0.001)		
Jalali et al. 2020 [203]	6 publications [34,50,161,164,165,177]	NAFLD	407	33–58	P	8–12	50–1425 mg/day of curcumin	↔ TG (MD = −0.608; 95% CI −1.253 to 0.038; *p* = 0.065) ↓ LDL-C (MD = −1.028; 95% CI −1.942 to −0.113; *p* = 0.028) ↓ TC (MD = −0.645; 95% CI −1.047 to −0.243; *p* = 0.002) ↔ HDL-C (MD = 0.880; 95% CI −0.285 to 2.044; *p* = 0.139)	Jadad scale (88.9%)	PubMed, Embase, Scopus, Web of Science, and Cochrane Library
Mendia et al. 2019 [213]	17 publications [45,51,52,65,101,114,118,126,138,161,177,182,183,214,215,216,217]	T2DM, Mets, NAFLD, COPD, dyslipidemia, CAD, acute coronary syndrome, healthy, elderly	1205	30–83	P, CO	4–24	45–4000 mg/day of curcumin, 294–1000 mg/day of curcuminoids, 1200 mg/day of turmeric	↔ LDL-C (WMD = −5.82 mg/dL; 95% CI −15.80 to 4.16; *p* = 0.253)	Cochrane risk of bias tool (55.0%)	PubMed–Medline, Scopus, and Web of Science
	19 publications [31,45,51,52,62,65,101,114,126,138,161,177,182,183,214,215,216,217,218]	T2DM, Mets, NAFLD, COPD, dyslipidemia, CAD, acute coronary syndrome, healthy, elderly	1439	23–83	P, CO	1–24	45–6000 mg/day of curcumin, 294–1000 mg/day of curcuminoids, 1200 mg/day of turmeric	↓ TG (WMD = −21.36 mg/dL; 95% CI −32.18 to −10.53; *p* < 0.001)		
	18 publications [45,51,52,62,101,114,118,126,138,161,177,182,183,214,215,216,217,218]	T2DM, Mets, NAFLD, COPD, dyslipidemia, CAD, acute coronary syndrome, healthy, elderly	1247	23–83	P, CO	1–24	45–6000 mg/day of curcumin, 294–1000 mg/day of curcuminoids, 1200 mg/day of turmeric	↔ TC (WMD = −9.57 mg/dL; 95% CI −20.89 to 1.75; *p* = 0.098)		
	16 publications [45,51,52,65,101,114,126,138,161,177,182,183,214,215,216,217]	T2DM, Mets, NAFLD, COPD, dyslipidemia, CAD, acute coronary syndrome, elderly	1145	30–83	P, CO	4–24	45–4000 mg/day of curcumin, 294–1000 mg/day of curcuminoids, 1200 mg/day of turmeric	↓ HDL-C levels (WMD = 1.42 mg/dL; 95% CI 0.03 to 2.81; *p* = 0.046)		
Azhdari et al. 2019 [204]	5 publications [25,61,126,138,172]	Mets	259	30–72	P	6–12	20–1890 mg/day of curcumin, 1000 mg/day of curcuminoids, 2400 mg/day of turmeric	↓ TG (WMD = −33.65 mg/dL; 95% CI −51.27 to −16.03; *p* < 0.001) ↑ HDL-C (WMD = 4.31 mg/dL; 95% CI 1.50 to 7.11; *p* = 0.003)	Cochrane risk of bias tool (59.2%)	Web of Science, MEDLINE, Google Scholar, Scopus, Cochrane, and CINAHL
Tabrizi et al. 2018 [207]	19 publications [25,31,45,51,52,57,87,101,114,126,138,161,178,183,206,208,214,219,220]	Mets, hyperlipidemia, T2DM, acute coronary syndrome, overweight/obese, diabetic nephropathy, NAFLD, CAD	1396	18–73	P	4–24	80–2000 mg/day of curcumin, 300–1000 mg/day of curcuminoids, 112–2100 mg/day of turmeric	↓ TG (SMD = −1.21; 95% CI −1.78 to −0.65; *p* < 0.001)	Cochrane risk of bias tool (76.9%)	Cochrane Library, EMBASE, MEDLINE, and Web of Science
	20 publications [25,45,51,52,57,87,101,114,126,138,161,178,182,183,206,208,209,214,219,220]	Mets, hyperlipidemia, T2DM, acute coronary syndrome, overweight/obese, diabetic nephropathy, NAFLD, CAD	1229	18–73	P	4–24	70–2000 mg/day of curcumin, 300–1000 mg/day of curcuminoids, 300–2100 mg/day of turmeric	↓ TC (SMD = −0.73; 95% CI −1.32 to −0.13; *p* = 0.01)		
	21 publications [25,45,46,51,52,57,87,101,114,126,138,161,178,182,183,206,208,209,214,219,220]	Mets, hyperlipidemia, T2DM, acute coronary syndrome, overweight/obese, diabetic nephropathy, NAFLD, CAD	1289	18–73	P	4–24	70–2000 mg/day of curcumin, 300–1000 mg/day of curcuminoids, 112–2100 mg/day of turmeric	↔ LDL-C (SMD = −0.52; 95% CI −1.14 to 0.11; *p* = 0.10)		
	20 publications [25,45,51,52,57,87,101,114,126,138,161,178,182,183,206,208,209,214,219,220]	Mets, hyperlipidemia, T2DM, acute coronary syndrome, overweight/obese, diabetic nephropathy, NAFLD, CAD	1229	18–73	P	4–24	70–2000 mg/day of curcumin, 300–1000 mg/day of curcuminoids, 300–2100 mg/day of turmeric	↔ HDL-C (SMD = 0.28; 95% CI −0.22 to 0.77; *p* = 0.27)		
Qin et al. 2017 [221]	7 publications [25,31,57,114,126,161,182]	T2DM, Mets, NAFLD, dyslipidemia, prediabetes, prehypertension	649	28–72	P	4–12	80 mg/day of curcumin, 70–1890 mg/day of curcuminoids, 2000–2400 mg/day of turmeric	↓ TG (SMD = −0.214; 95% CI −0.369 to −0.059; *p* = 0.007)	Cochrane risk of bias tool (57.1%)	PubMed, Embase, Ovid, Medlin, and Cochrane Library
	6 publications [25,31,57,126,161,182]	T2DM, Mets, NAFLD, dyslipidemia, prediabetes, prehypertension	512	28–72	P	4–12	80 mg/day of curcumin, 70–1890 mg/day of curcuminoids, 2000–2400 mg/day of turmeric	↓ LDL-C (SMD = −0.340; 95% CI −0.530 to −0.150; *p* < 0.0001) ↔ TC (SMD = −0.38; 95% CI −0.78 to 0.01; *p* = 0.054) ↔ HDL-C (SMD = 0.09; 95% CI −0.10 to 0.27; *p* = 0.370)		
Sahebkar et al. 2014 [222]	5 publications [52,114,214,215,218]	T2DM, healthy obese, dyslipidemia, acute coronary syndrome, Alzheimer’s disease	197	23–83	P, CO	1–24	45–6000 mg/day of curcuminoids	↔ TG (MD = −1.29 mg/dL; 95% CI −9.05 to 6.48; *p* = 0.75) ↔ TC (MD = 8.97 mg/dL; 95% CI −4.56 to 22.51; *p* = 0.19)	Cochrane risk of bias tool (33.3%)	PubMed–Medline, SCOPUS, Ovid-AMED, ClinicalTrials.gov, and Cochrane Library
	4 publications [52,114,214,215]	T2DM, obese, dyslipidemia, acute coronary syndrome, Alzheimer’s disease	173	27–83	P, CO	8–24	45–4000 mg/day of curcuminoids	↔ LDL-C (MD = 16.15 mg/dL; 95% CI −4.43 to 36.74; *p* = 0.12) ↔ HDL-C (MD = −0.59 mg/dL; 95% CI −1.66 to 0.49; *p* = 0.28)		

CAD, coronary artery disease; CO, crossover; COPD, chronic obstructive pulmonary disease; CI, confidence interval; CKD, chronic kidney disease; CVD, cardiovascular disease; DFU, diabetic foot ulcer; ES, effect size; ESRD, end-stage renal disease; HDL, high-density lipoprotein; HDL-C, high-density lipoprotein cholesterol; IFG, impaired fasting glucose; IGT, impaired glucose tolerance; LDL, low-density lipoprotein; LDL-C, low-density lipoprotein cholesterol; MAFLD, metabolic (dysfunction)-associated fatty liver disease; MD, mean difference; Mets, metabolic syndrome; NAFLD, non-alcoholic fatty liver disease; P, parallel; PCOS, polycystic ovary syndrome; RCTs, randomized controlled trials; TC, total cholesterol; T2DM, type 2 diabetes mellitus; TG, triglyceride; WMD, weighted mean difference. The quality of primary studies is indicated as a percentage of low risk (Cochrane risk of bias tool) or ≥3 (Jadad scale). ↑, increase; ↓, decrease; ↔ no effect.

**Table 5 nutrients-16-01728-t005:** Meta-analyses of RCTs investigating the effect of curcumin intake on body weight, BMI, and waist circumference.

Ref.	No. of Studies Included	Health Status of Subjects	No. of Subjects	Age of Subjects (Years)	Design	Period (Weeks)	Dose	Outcomes (Effect Size)	Quality of Primary Studies	Databases
Lukkunaprasit et al. 2023 [160]	13 publications [34,35,50,161,164,165,166,167,168,175,176,178,181]	MAFLD	756	25–38	P	8–12	80–1500 mg/day of curcumin, 3000 mg/day of turmeric	↓ BMI (MD = −0.34; 95% CI −0.62 to −0.05; *p* < 0.05)	Cochrane risk of bias tool (68.8%)	Medline and SCOPUS
Qiu et al. 2023 [22]	7 publications [23,25,61,170,171,172,223]	Mets	369	28–64	P	4–12	80–1000 mg/day of curcumin, 2400 mg/day of turmeric	↓ WC (MD = −2.16; 95% CI −3.78 to −0.54; *p* = 0.009)	Cochrane risk of bias tool (71.4%)	PubMed, SCOPUS, Cochrane Library, EMBASE, Web of Science, and China Biological Medicine
Rozanski et al. 2023 [173]	14 publications [34,35,50,55,161,164,165,166,174,175,176,177,178,179]	MAFLD	847	18–70	P	8–12	70–1500 mg/day of curcuminoids, 80–1000 mg/day of curcumin, 3000 mg/day of turmeric	↓ WC (MD = −2.6303; 95% CI −4.9350 to −0.3256; *p* = 0.0253)	Cochrane risk of bias tool (90.8%)	PubMed, Web of Science, and Scopus
Ngu et al. 2022 [180]	11 publications [34,35,50,161,164,166,175,178,181,196,224]	NAFLD	701	18–70	P	8–12	50–1500 mg/day of curcumin, 2000 mg/day of turmeric	↓ BMI (MD = −0.41; 95% CI −0.75 to − 0.07; *p* = 0.02)	Cochrane risk of bias tool (79.5%)	Cochrane Central Register of Controlled Trials and PubMed
Sun et al. 2022 [100]	15 publications [23,25,38,49,60,61,70,101,126,170,182,183,185,209,223]	Mets, obese, hyperlipidemia, hypercholesterolemia, T2DM, IGT, IFG, DSPN	990	38–62	P, CO	4–12	80–2400 mg/day of curcumin	↓ BW (WMD = −0.94 kg; 95% CI −1.40 to −0.47; *p* < 0.001)	Cochrane risk of bias tool (69.3%) Jadad scale (77.0%)	PubMed, Embase, Web of Science, CNKI, Wanfang, and CBM
	12 publications [23,25,38,49,61,70,170,183,185,188,209,223]	Mets, obese, hypercholesterolemia, T2DM, IGT, IFG, DSPN	882	38–63	P, CO	4–17	80–2400 mg/day of curcumin	↔ WC (WMD = −1.41 cm; 95% CI −2.82 to 0.004; *p* = 0.051)		
	16 publications [23,25,29,49,60,61,76,101,126,170,182,183,185,188,209,223]	Mets, obese, hyperlipidemia, hypercholesterolemia, T2DM, IGT, IFG, DSPN, hemodialysis	1135	38–70	P, CO	4–24	80–2400 mg/day of curcumin	↓ BMI (WMD = −0.40 kg/m^2^; 95% CI −0.60 to −0.19; *p* < 0.001)		
Khalili et al. 2022 [195]	9 publications [34,50,161,165,166,176,179,181,196]	NAFLD	619	42–67	P	4–12	80–2000 mg/day of curcumin	↔ BMI (SMD = −0.13; 95% CI −0.29 to 0.02; *p* = 0.096)	Cochrane risk of bias tool (65.5%)	Web of Science, Science Direct, PubMed, and Embase
Nouri et al. 2022 [192]	4 publications [58,73,193,194]	PCOS	198	28–31	P	6–12	93.34–1500 mg/day of curcumin	↔ BMI (ES = −0.23 kg/m^2^; 95% CI −1.46 to 0.99; *p* = 0.70)	Cochrane risk of bias tool (75.0%)	PubMed, Scopus, and ISI Web of Science
Shen et al. 2022 [99]	7 publications [58,73,193,194,197,198,199]	PCOS	447	20–38	P	6–24	80–1500 mg/day of curcumin	↓ BMI (WMD = −0.267; 95% CI −0.450 to −0.084; *p* = 0.004)	Cochrane risk of bias tool (73.5%)	PubMed, Embase, Cochrane Library, Web of Science, Scopus, Clinical Trials, Chinese Clinical Trial Registry, Chinese Biomedical Literature Database, Chinese National Knowledge Infrastructure, VIP database, and Wanfang Database
	4 publications [58,73,193,197]	PCOS	229	24–35	P	6–12	93.34–1000 mg/day of curcumin	↔ BW (WMD = −0.924; 95% CI −2.009 to 0.162; *p* = 0.095)		
	2 publications [73,194]	PCOS	97	24–35	P	8–12	93.34–1500 mg/day of curcumin	↔ WC (WMD = −1.475; 95% CI −4.519 to 1.570; *p* = 0.342)		
Abdelazeem et al. 2021 [115]	4 publications [58,73,193,197]	PCOS	229	24–35	P	6–12	80–1000 mg/day of curcumin	↔ BW (MD = −0.15; 95% CI −0.56 to 0.27; *p* = 0.49)	Cochrane risk of bias tool (73.3%)	PubMed, EMBASE, Scopus, Web of Science, Cochrane Central, and Google Scholar
	2 publications [73,194]	PCOS	97	24–35	P	8–12	93.34–1500 mg/day of curcumin	↔ WC (MD = −1.47; 95% CI −4.52 to 1.57; *p* = 0.34)		
	5 publications [58,73,193,194,197]	PCOS	296	24–39	P	6–12	80–1500 mg/day of curcumin	↔ BMI (MD = −0.15; 95% CI −0.56 to 0.27; *p* = 0.49)		
Altobelli et al. 2021 [201]	3 publications [49,144,182]	T2DM	168	30–70	P	8–12	80–1500 mg/day of curcumin, 2100 mg/day of turmeric	↔ BMI (ES = −0.30 kg/m^2^; 95% CI −0.62 to 0.02; *p* = 0.067)	Cochrane risk of bias tool (93.8%)	Medline, EMBASE, ClinicalTrials.gov, Web of Science, Cochrane Library, and Scopus
Ashtary-Larky et al. 2021 [111]	4 publications [23,50,171,185]	Mets, T2DM, NAFLD	240	36–64	P	6–12	40–80 mg/day of nano-curcumin	↔ BW (WMD = −0.51 mg/dL; 95% CI −1.85 to 0.82; *p* = 0.449)	Cochrane risk of bias tool (55.6%)	PubMed, Scopus, Embase, and Web of Science
	6 publications [23,50,71,171,182,185]	Mets, T2DM, NAFLD	364	36–67	P	6–12	40–80 mg/day of nano-curcumin	↔ BMI (WMD = −0.35 mg/dL; 95% CI −0.76 to 0.04; *p* = 0.079)		
	4 publications [23,50,171,185]	Mets, T2DM, NAFLD	240	36–64	P	6–12	40–80 mg/day of nano-curcumin	↔ WC (WMD = −1.32 mg/dL; 95% CI −3.89 to 1.23; *p* = 0.310)		
	2 publications [23,171]	Mets, NAFLD	160	36–64	P	6–12	40–80 mg/day of nano-curcumin	↔ FM (WMD = −0.86 mg/dL; 95% CI −1.95 to 0.23; *p* = 0.123)		
Mousavi et al. 2020 [225]	8 publications [39,52,124,126,161,223,226,227]	Mets, prediabetic, obese, NAFLD, T2DM	667	38–59	P	4–39	80–200 mg/day of curcumin, 1000–1500 mg/day of curcuminoids	↓ BW (WMD = −1.14 kg; 95% CI −2.16 to −0.12; *p* = 0.02)	Jadad scale (81.8%)	PubMed, Medline, SCOPUS, Cochrane Library. and Google Scholar
	10 publications [52,89,124,126,161,178,182,223,226,227]	Mets, obese, NAFLD, T2DM	645	26–59	P	4–13	80–1000 mg/day of curcumin, 1000–1900 mg/day of curcuminoids	↓ BMI (WMD = −0.48 kg/m^2^; 95% CI −0.78 to −0.17; *p* = 0.002)		
	6 publications [39,52,89,178,223,226]	Mets, prediabetic, obese, NAFLD	456	26–57	P	4–39	200–1000 mg/day of curcumin, 1000–1500 mg/day of curcuminoids	↔ WC (WMD = −1.51 cm; 95% CI −4.041 to 1.003; *p* = 0.23)		
Jalali et al. 2020 [203]	5 publications [34,50,161,164,165]	NAFLD	270	28–59	P	8–12	50–80 mg/day of curcumin	↔ BW (MD = −0.051; 95% CI −0.271 to 0.170; *p* = 0.653)	Jadad scale (88.9%)	PubMed, Embase, Scopus, Web of Science. and Cochrane Library
	7 publications [34,50,124,161,164,165,181]	NAFLD	453	28–59	P	8–12	50–1500 mg/day of curcumin	↔ BMI (MD = −0.179; 95% CI −0.365 to 0.006; *p* = 0.058)		
	3 publications [50,124,165]	NAFLD	269	35–58	P	8–12	80–1425 mg/day of curcumin	↓ WC (MD = −1.005; 95% CI −1.304 to −0.706; *p* < 0.0001)		
Baziar et al. 2020 [228]	6 publications [34,35,50,161,164,165]	NAFLD	385	42–46	P	8–12	70–1500 mg/day of curcumin, 3000 mg/day of turmeric	↔ BW (WMD = −0.60 kg; 95% CI −2.07 to 0.87; *p* = 0.423)	Grading of Recommendations, Assessment, Development, and Evaluation (GRADE) (87.5%)	PubMed/Medline, ISI Web of Science, Scopus, EMBASE, Cochrane Library, Google Scholar, Sid.ir, and Magiran.com
	8 publications [34,35,50,161,164,165,178,179]	NAFLD	567	42–54	P	8–12	70–1500 mg/day of curcumin, 3000 mg/day of turmeric	↓ BMI (WMD = −0.34 kg/m^2^; 95% CI −0.64 to −0.0; *p* < 0.05)		
	4 publications [50,165,178,179]	NAFLD	318	42–54	P	8–12	70–1500 mg/day of curcumin, 3000 mg/day of turmeric	↓ WC (WMD = −2.12 cm; 95% CI −3.26 to −0.98; *p* < 0.001)		
Akbari et al. 2019 [131]	12 publications [25,45,52,124,126,161,178,182,208,223,226,229]	Mets, obese, T2DM, hyperlipidemia, NAFLD	854	23–72	P	4–12	80–2400 mg/day of curcumin, 294 mg/day of curcuminoids, 2100 mg/day of turmeric	↓ BMI (SMD = −0.37; 95% CI −0.61 to 0.13; *p* < 0.01)	Cochrane risk of bias tool (77.8%)	Cochrane Library, EMBASE, PubMed, and Web of Science
	13 publications [25,32,39,45,46,52,126,161,178,208,223,226,229]	Mets, obese, T2DM, hyperlipidemia, NAFLD	1233	10–72	P	4–36	80–2400 mg/day of curcumin, 294–1500 mg/day of curcuminoids, 2100–2800 mg/day of turmeric	↓ BW (SMD = −0.23; 95% CI −0.39 to 0.06; *p* < 0.01)		
	8 publications [25,31,32,39,52,124,223,226]	Mets, obese, T2DM	848	10–72	P	4–36	200–2400 mg/day of curcumin, 1500 mg/day of curcuminoids	↓ WC (SMD = −0.25; 95% CI −0.44 to −0.05; *p* = 0.01)		
	5 publications [25,32,52,223,226]	Mets, obese	301	10–58	P	4–8	200–2400 mg/day of curcumin	↔ hip ratio (SMD = −0.17; 95% CI −0.42 to 0.08; *p* = 0.18)		
Wei et al. 2019 [230]	2 publications [161,231]	NAFLD	299	41–57	P	8–24	500–3000 mg/day of curcumin	↓ BW (MD = −2.27; 95% CI −3.11 to −1.44; *p* < 0.0001)	Cochrane risk of bias tool (75.0%)	PubMed, EMBASE, and Cochrane Library
Azhdari et al. 2019 [204]	4 publications [25,61,172,223]	Mets	244	38–48	P	4–8	800–2400 mg/day of curcumin	↔ WC (WMD = −0.41 cm; 95% CI −1.95 to 1.13; *p* = 0.6)	Cochrane risk of bias tool (59.2%)	Web of Science, MEDLINE, Google Scholar, Scopus, Cochrane, and CINAHL
Jafarirad et al. 2019 [232]	7 publications [35,146,161,165,177,227,233]	NAFLD	429	40–65	P	8–12	80–1500 mg/day of curcumin, 2000–3000 mg/day of turmeric	↔ BMI (WMD = −0.21 kg/m^2^; 95% CI −0.71 to 0.28; *p* = 0.39)	Cochrane risk of bias tool (72.9%)	PubMed, Scopus, Cochrane Library, and ISI Web of Science
	7 publications [35,146,161,163,165,227,233]	NAFLD	362	40–65	P	8–24	80–1500 mg/day of curcumin, 3000 mg/day of turmeric	↔ BW (WMD = −0.54 kg; 95% CI −2.40 to 1.31; *p* = 0.56)		
	4 publications [146,163,165,177]	NAFLD	221	40–54	P	8–24	1000–1500 mg/day of curcumin, 2000 mg/day of turmeric	↔ WC (WMD = −0.88 cm; 95% CI −3.76 to 2.00; *p* = 0.54)		

BMI, body mass index; BW, body weight; CI, confidence interval; CO, crossover; DSPN, diabetic sensorimotor polyneuropathy; ES, effect size; FM, fat mass; IFG, impaired fasting glucose; IGT, impaired glucose tolerance; MAFLD, metabolic (dysfunction)-associated fatty liver disease; MD, mean difference; Mets, metabolic syndrome; NAFLD, nonalcoholic fatty liver disease; P, parallel; PCOS, polycystic ovary syndrome; RCTs, randomized controlled trials; SMD, standardized mean difference; T2DM, type 2 diabetes mellitus; WC, waist circumference; WMD, weighted mean differences. The quality of primary studies is indicated as a percentage of low risk (Cochrane risk of bias tool) or ≥3 (Jadad scale) or good (GRADE). ↑, increase; ↓, decrease; ↔ no effect.

**Table 6 nutrients-16-01728-t006:** Meta-analyses of RCTs investigating the effect of curcumin intake on blood pressure and endothelial function.

Ref.	No. of Studies Included	Health Status of Subjects	No. of Subjects	Age of Subjects (Years)	Design	Period (Weeks)	Dose	Outcomes (Effect Size)	Quality of Primary Studies	Databases
Lukkunaprasit et al. 2023 [160]	5 publications [50,166,167,168,178]	MAFLD	358	25–36	P	8–12	80–1000 mg/day of curcumin	↔ SBP (MD = −0.29; 95% CI −0.91 to 0.34; *p* = 0.37) ↔ DBP (MD = −0.02; 95% CI −0.55 to 0.52; *p* = 0.95)	Cochrane risk of bias tool (68.8%)	Medline and SCOPUS
Qiu et al. 2023 [22]	6 publications [23,25,27,61,170,171]	Mets	405	28–61	P	6–12	80–1000 mg/day of curcumin, 1000 mg/day of curcuminoids, 2400 mg/day of turmeric	↔ SBP (MD = −4.82; 95% CI −9.98 to 0.35; *p* = 0.07)	Cochrane risk of bias tool (71.4%)	PubMed, SCOPUS, Cochrane Library, EMBASE, Web of Science, and China Biological Medicine
	5 publications [25,27,61,170,171]	Mets	383	28–61	P	6–12	80–1000 mg/day of curcumin, 1000 mg/day of curcuminoids, 2400 mg/day of turmeric	↓ DBP (MD = −2.8; 95% CI −4.53 to −1.06; *p* = 0.002)		
Ashtary-Larky et al. 2021 [111]	4 publications [23,50,76,171]	Mets, NAFLD, T2DM, hemodialysis	213	37–68	P	6–12	40–80 mg/day of nano-curcumin	↓ SBP (WMD = −7.09 mg/dL; 95% CI −12.98 to −1.20; *p* < 0.001)	Cochrane risk of bias tool (55.6%)	PubMed, Scopus, Embase, and Web of Science
	3 publications [50,76,171]	Mets, NAFLD, T2DM, hemodialysis	180	37–68	P	12	40–80 mg/day of nano-curcumin,	↔ DBP (WMD = −0.07 mg/dL; 95% CI −1.12 to 0.97; *p* = 0.891)		
Changal et al. 2020 [234]	5 publications [66,235,236,237,238]	Healthy, postmenopausal	733	19–70	P	One time, 8–12	25–5000 mg/day of curcumin	↑ FMD (SDM = 1.379; 95% CI 0.48 to 2.274; *p* = 0.003)	Not assessed	PubMed, ClinicalTrials.gov, Scopus, and Embase
Hallajzadeh et al. 2019 [239]	2 publications [66,235]	Healthy	60	59–71	P	8–12	150–2000 mg/day of curcumin	↑ FMD (WMD = 1.49; 95% CI 0.16 to 2.82; *p* < 0.05)	Cochrane risk of bias tool (not indicated)	Cochrane Library, EMBASE, PubMed, and Web of Science
	4 publications [31,66,89,240]	Healthy, postmenopausal, obese, T2DM	319	22–70	P	8–24	150–2000 mg/day of curcumin	↔ PWV (WMD = −41.59; 95% CI −86.59 to 3.42; *p* > 0.05)		
	2 publications [66,240]	Healthy, postmenopausal	84	53–68	P	8–12	150–2000 mg/day of curcumin	↔ Aix (WMD = 0.71; 95% CI −1.37 to 2.79; *p* > 0.05)		
	3 publications [33,66,114]	Healthy, depressive, T2DM	133	31–67	P	8–12	150–2000 mg/day of curcumin	↔ ET-1 (WMD = −0.30; 95% CI −0.96 to 0.37; *p* > 0.05)		
	3 publications [62,120,241]	Healthy, perennial allergic rhinitis, dyslipidemia	353	26–62	P	4–12	80–500 mg/day of curcumin	↔ sICAM-1 (WMD = −10.11; 95% CI −33.67 to 13.46; *p* > 0.05)		
Hadi et al. 2019 [242]	11 publications [25,50,61,65,66,89,151,178,219,243,244]	Mets, obese, lupus nephritis, T2DM postmenopausal, CKD, NAFLD, COPD, diabetic proteinuria, healthy, elderly	734	21–74	P, CO	8–24	80–1000 mg/day of curcumin, 320–2400 mg/day of turmeric	↔ SBP (ES = −0.69 mmHg; 95% CI −2.01 to 0.64) ↔ DBP (ES = 0.28 mmHg; 95% CI −1.12 to 1.68)	Cochrane risk of bias tool (69.7%)	Medline, EMBASE, SCOPUS, ISI Web of Science, Google Scholar, and Cochrane Library
Azhdari et al. 2019 [204]	3 publications [25,27,61]	Mets	280	38–45	P	8	1000 mg/day of curcumin, 1000 mg/day of curcuminoids, 2400 mg/day of turmeric	↔ SBP (WMD = −1.68 mmHg; 95% CI −4.68 to 1.3) ↓ DBP (WMD = −2.96 mmHg; 95% CI −5.09 to −0.83; *p* = 0.007)	Cochrane risk of bias tool (59.2%)	Web of Science, MEDLINE, Google Scholar, Scopus, Cochrane, and CINAHL

Aix, augmentation index; CI, confidence interval; CKD, chronic kidney disease; COPD, chronic obstructive pulmonary disease; DBP, diastolic blood pressure; ES, effect size; ET-1, endothelin-1; FMD, flow-mediated dilation; MAFLD, metabolic (dysfunction)-associated fatty liver disease; MD, mean difference; Mets, metabolic syndrome; NAFLD, nonalcoholic fatty liver disease; P, parallel; PWV, pulse wave velocity; RCTs, randomized controlled trials; SBP, systolic blood pressure; SDM, standard difference in means; sICAM-1, soluble intercellular adhesion molecule-1; T2DM, type 2 diabetes mellitus; WMD weighted mean differences. The quality of primary studies is indicated as a percentage of low risk (Cochrane risk of bias tool) or ≥3 (Jadad scale). ↑, increase; ↓, decrease; ↔ no effect.

**Table 7 nutrients-16-01728-t007:** Meta-analyses of RCTs investigating the effect of curcumin intake on depression and cognitive function.

Ref.	No. of Studies Included	Health Status of Subjects	No. of Subjects	Age of Subjects (Years)	Design	Period (Weeks)	Dose	Outcomes (Effect Size)	Quality of Primary Studies	Databases
Wang et al. 2021 [245]	6 publications [139,246,247,248,249,250]	MDD (DSM-IV or Mini 6.0)	520	40–76	P	5–12	500–1500 mg/day of curcumin	↓ Depression (SMD = −0.35; 95% CI −0.56 to −0.15; I^2^ = 7%)	Cochrane risk of bias tool (71.7%)	EMBASE, PubMed, PsycINFO, Web of Science, Cochrane Library, and ClinicalTrials.gov
	4 publications [251,252,253,254]	T2DM, obesity, schizophrenia (DSM-IV), systemic lupus erythematosus	77	25–68	P, CO	4–24	80–3000 mg/day of curcumin	↔ Depressive symptoms (SMD = −0.32; 95% CI −0.50 to −0.13; I^2^ = 15%)		
	3 publications [139,248,250]	MDD (DSM-IV)	440	23–53	P	6–12	500–1000 mg/day of curcumin	↑ Response rates (OR = 3.20; 95% CI 1.28 to 7.99; I^2^ = 35%)		
Fusar-Poli et al. 2020 [255]	9 publications [139,246,247,248,249,250,252,254,256]	MDD (DSM-IV), obesity, systemic lupus erythematosus	599	17–81	P, CO	4–12	150–1500 mg/day of curcumin	↓ Depression (Hedge’s g = −0.75; 95% CI −1.11 to −0.39; *p* < 0.001; I^2^ = 26.28%)	Cochrane risk of bias tool (71.4%)	Web of Science, MEDLINE, KCI, Russian Science Citation Index, SciELO Citation Index, CINAHL, Embase, PsycINFO, and ClinicalTrials.gov
	4 publications [248,249,252,256]	MDD (DSM-IV), obesity	320	17–80	P	4–12	500–1000 mg/day of curcumin	↓ Anxiety symptoms (Hedge’s g = −2.62; 95% CI −4.06 to −1.17; *p* < 0.001)		
unNg et al. 2017 [257]	6 publications [139,246,249,250,252,256]	CGI-S ≥ 4, HAM-D_17_ ≥ 21, MADRS ≥ 22, BDI-II, DSM-IV, IDS-SR_30_ ≥ 14, HAM-D_17_ ≥ 7, HAM-D_17_ ≥ 10, MADRS ≥ 14	377	30–76	P, CO	4–8	500–1000 mg/day of curcumin	↓ Depressive symptoms (SMD = −0.344; 95% CI −0.558 to −0.129; *p* = 0.002)	Jadad scale (100%)	PubMed, Ovid, Clinical Trials Register of the Cochrane Collaboration Depression, Anxiety, and Neurosis Group (CCDANTR), and Cochrane Field for Complementary Medicine
Al-Karawi et al. 2017 [258]	6 publications [139,246,249,250,252,256]	CGI-S ≥ 4, HAM-D_17_ ≥ 21, MADRS ≥ 22, BDI, DSM-IV, IDS-SR_30_ ≥ 14, HAM-D_17_ ≥ 7 HAM-D_17_ ≥ 10, MADRS ≥ 14	377	30–76	P, CO	4–8	500–1000 mg/day of curcumin	↓ Major depression (SMD = −0.34; 95% CI −0.56 to −0.13; *p* = 0.002)	Quality Assessment Tool for Quantitative Studies (83.3%)	Pubmed, Scopus, Psychinfo, Evidence-Based Medicine Guidelines, DynaMed, JAMA evidence, and Cochrane Library
Tsai et al. 2021 [259]	6 publications [260,261,262,263,264,265]	Alzheimer’s disease, obese, schizophrenia, healthy older adults	264	28–82	P	6–24	160–4000 mg/day of curcumin	↔ Cognitive function (Hedges’ g = 0.340; 95% CI −0.353 to 1.033; *p* = 0.337; I^2^ = 0.0%)	Cochrane risk of bias tool (89.6%)	PubMed, Embase, ClinicalKey, Cochrane CENTRAL, ProQuest, ScienceDirect, and Web of Science
	3 publications [261,266,267]	Schizophrenia, obese, healthy older adults	160	28–76	P	8–16	80–180 mg/day of curcumin	↑ Working memory (Hedges’ g = 0.396; 95% CI 0.078 to 0.714; *p* = 0.015; I^2^ = 0.0%)		
Zhu et al. 2019 [268]	3 publications [118,263,269]	healthy, elderly, nondemented	196	54–73	P	4–72	80–1320 mg/day of curcumin	↑ Cognitive function in the elderly (SMD = 0.33; 95% CI 0.05 to 0.62; *p* = 0.02)	Cochrane risk of bias tool (80.0%)	PubMed, Embase, Web of Science, ClinicalTrials.gov, Cochrane Library, Chinese National Knowledge Infrastructure, Wanfang Data, and China Biology Medicine disc
Sarraf et al. 2019 [270]	4 publications [172,262,271,272]	Mets, premenstrual syndrome, schizophrenia,	139	18–65	P	8–12	200–1820 mg/day of curcumin	↑ BDNF (WMD = 1789.38 pg/mL; 95% CI 722.04 to 2856.71; *p* < 0.01; I^2^ = 83.5%; *p* < 0.001)	Cochrane risk of bias tool (not indicated)	PubMed, Scopus, Web of Science, Cochrane Library, and Google Scholar

BDNF, brain-derived neurotrophic factor; BDI-II, Beck Depression Inventory II; CGI-S, Clinical Global Impression Severity Scale; CI, confidence interval; CO, crossover; DSM, diagnostic and statistical manual of mental disorders; HAM-D_17_, Hamilton Depression Rating Scale, 17-item version; IDS-SR_30_, Inventory of Depressive Symptomatology—Self-rated; MADRS, Montgomery–Åsberg Depression Rating Scale; MDD, major depressive disorder; Mets, metabolic syndrome; OR, odds ratio; P, parallel; RCTs, randomized controlled trials; SMD, standardized mean difference; T2DM, type 2 diabetes mellitus; WMD, weighted mean differences. The quality of primary studies is indicated as a percentage of low risk (Cochrane risk of bias tool) or ≥3 (Jadad scale). ↑, increase; ↓, decrease; ↔ no effect.

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
