# Peer review of "Is Curcumin Intake Really Effective for Chronic Inflammatory Metabolic Disease? A Review of Meta-Analyses of Randomized Controlled Trials"

_nutrients, 2024, doi:10.3390/nu16111728_

Round 1
Reviewer 1 Report
Comments and Suggestions for Authors
Comments:
The manuscript aimed to examine the effects of curcumin on chronic inflammatory metabolic disease by extensively evaluating meta-analyses of randomized controlled trials (RCTs). It is an interesting study, but I have some considerations that should be taken into account to improve the quality of the paper:
1. The introduction is too short to attract readers following your topic. Suggest add 1-2 paragraphs to illustrate the significance of your review, why do the research between chronic disease and curcumin.
2. What is the innovation of your research, fill it in the introduction.
3. Line 85-87, what is the reason for Records excluded
4. For results, add concentrations when comparing the effects of curcumin, especially those reports without activities.
5. With regard to discussion, please raise a hypothesis to explain why curcumin can not show significant effects to chronic disease in those publications?
6. Update your citations with some publications in 2024.
7. For conclusion, in my opinion, the dose of curcumin should be mentioned as a highlight of your conclusion.
Author Response
The manuscript aimed to examine the effects of curcumin on chronic inflammatory metabolic disease by extensively evaluating meta-analyses of randomized controlled trials (RCTs). It is an interesting study, but I have some considerations that should be taken into account to improve the quality of the paper:
- The introduction is too short to attract readers following your topic. Suggest add 1-2 paragraphs to illustrate the significance of your review, why do the research between chronic disease and curcumin.
Thank you. We have revised introduction as you suggested. Please refer to lines 41-46.
- What is the innovation of your research, fill it in the introduction.
Thank you. We have highlighted the innovation of our research as you suggested. Please refer to lines 57-59, 62-64.
- Line 85-87, what is the reason for Records excluded
Thank you. We have added the reason for removed studies after screening the titles and abstracts as you suggested. Please refer to Figure 1.
- For results, add concentrations when comparing the effects of curcumin, especially those reports without activities.
Thank you. We have revised through the manuscript by adding concentrations when comparing the effects of curcumin, especially those reports without activities.
- With regard to discussion, please raise a hypothesis to explain why curcumin cannot show significant effects to chronic disease in those publications?
Thank you. We have added additional discussion about insignificant effect of curcumin on anthropometric outcomes. Please refer to line 574-582.
- Update your citations with some publications in 2024
Thank you. We have updated our citations with some publications in 2024. Please refer to lines 45, 59.
- For conclusion, in my opinion, the dose of curcumin should be mentioned as a highlight of your conclusion.
Thank you. The doses of curcumin have been highlighted in lines 599-615 of conclusion.

Reviewer 2 Report
Comments and Suggestions for Authors
The Review study is complex and rigorously scientific.
The study has a high utility for pharmacological research and pharmaceutical chemistry as well; this provides a quick view of the doses and biological effects expected too.
In my opinion, an important contribution could be made through adding the PCA (Principal component analysis) technique in exploratory data analysis results, their visualization and preprocessing, applied at each one of the studied criteria (the glycemic control, glucose, lipids, blood pressure (BP), inflammatory markers, type 2 diabetes mellitus (T2DM), hypertension, metabolic syndrome, non-alcoholic fatty liver disease (NAFLD), cardiovascular disease (CVD), endothelial function, flow-mediated dilation (FMD), obesity, body weight (BW), depression and cognitive function).
Author Response
The Review study is complex and rigorously scientific.
The study has a high utility for pharmacological research and pharmaceutical chemistry as well; this provides a quick view of the doses and biological effects expected too.
In my opinion, an important contribution could be made through adding the PCA (Principal component analysis) technique in exploratory data analysis results, their visualization and preprocessing, applied at each one of the studied criteria (the glycemic control, glucose, lipids, blood pressure (BP), inflammatory markers, type 2 diabetes mellitus (T2DM), hypertension, metabolic syndrome, non-alcoholic fatty liver disease (NAFLD), cardiovascular disease (CVD), endothelial function, flow-mediated dilation (FMD), obesity, body weight (BW), depression and cognitive function).
Thank you for your great advice. We absolutely agree that adding the PCA technique in exploratory data analysis results in improvement of our paper. However, it is rather difficult for us to perform this analysis at our current level because it takes a lot of time and effort. We would like to apply it in future research.
